# Echoes as Anchors: Probabilistic Costs and Attention Refocusing in LLM Reasoning

**Zhuoyuan Hao**[1]  **Zhuo Li**[1]  **Wu Li**[1]  **Fangming Liu**[2,3]  **Min Zhang**[1]  **Jing Li**[1][✉]

[1]Harbin Institute of Technology, Shenzhen, China
[2]Pengcheng Laboratory, Shenzhen, China
[3]Huazhong University of Science and Technology, China
`hzy2210@gmail.com`  `jingli.phd@hotmail.com`

## Abstract

Test-time compute allocation in large reasoning models (LRMs) is widely used and has applications in mathematical problem solving, code synthesis, and planning. Recent work has addressed this problem by scaling self-consistency and parallel thinking, adding generic "thinking tokens" and prompting models to re-read the question before answering. Unfortunately, these approaches either inject task-agnostic tokens or mandate heuristics that do not explain—and often ignore—the *spontaneous* repetition that many LRMs exhibit at the head of their internal chains. In contrast, we analyze and harness the model's tendency to restate the question, which we term the *Echo of Prompt (EOP)*, as a front-loaded, compute-shaping mechanism. We formalize its probabilistic cost by casting echo removal as rejection-based conditioning and defining the *Echo Likelihood Gap* $\Delta\mathcal{L}$ as a computable proxy. This provides the missing theoretical link that links early repetition to likelihood gains and downstream accuracy. However, it does not by itself specify how to exploit EOP. Consequently, we develop *Echo-Distilled SFT (ED-SFT)* to instill an "echo-then-reason" pattern through supervised finetuning, and *Echoic Prompting (EP)* to re-ground the model mid-trace without training. While promising, quantifying benefits beyond verbosity is non-trivial. Therefore, we conduct length and suffix-controlled likelihood analyses together with layer-wise attention studies, showing that EOP increases answer to answer-prefix attention in middle layers, consistent with an *attention refocusing* mechanism. We evaluate on GSM8K, MathQA, Hendrycks-MATH, AIME24, and MATH-500 under identical decoding settings and budgets, and find consistent gains over baselines. Code is available at `https://github.com/hhh2210/echoes-as-anchors`.

## 1 Introduction

Recent advancements in Large Language Models (LLMs) have demonstrated remarkable capabilities in complex reasoning tasks, often mediated by a process known as Chain-of-Thought (CoT) prompting (Wei et al., 2022; Kojima et al., 2022; Wang et al., 2023; Yao et al., 2023).

Inspired by the CoT paradigm, modern large *reasoning* models (LRMs) achieve strong performance on complex tasks by allocating significant test-time compute to think before answering (Wei et al., 2022; OpenAI, 2024; DeepSeek-AI et al., 2025; Qwen Team, 2025). A common yet underexplored phenomenon in their reasoning traces is the tendency to begin by repeating the user's prompt (see Figure 1 and §A.3), a behavior we term the *Echo of Prompt* (EOP).

While uncontrolled repetition is a known failure mode (the "repeat curse" (Yao et al., 2025)), explicit instructions to re-read or "look-twice" are known to improve performance (Xu et al., 2024; Zou et al., 2024). The spontaneous EOP that initiates a complex reasoning trace, however, has remained largely unanalyzed.This initial echo raises a critical question:

*Is it a superfluous artifact of the training process, or does it serve a functional role in reasoning?*

---

✉ Corresponding author.

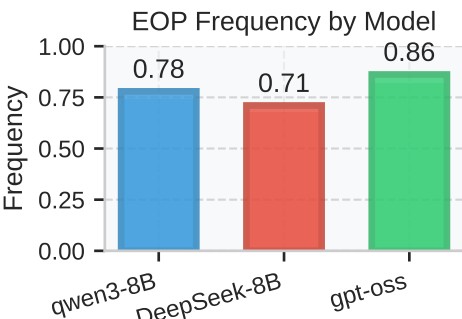

*User Query: Determine the radius of a cylindrical can given the area of a label... and the height...*
*Model's Echo: <think>Okay, let me see. The problem is asking for the radius of a cylindrical can. They give the area of the label as a quadratic expression..., and the height is also given...*

Figure 1: An illustration of the Echo of Prompt (EOP). Left: An example of a model's thinking process starting with an echo of the user's query. Right: The frequency of EOP across several open-source models on the GSM8K dataset, as measured by our trained MLP probe (see §A.4).

This paper confronts this question directly, presenting the first systematic study to isolate, analyze, and harness this emergent behavior as a powerful cognitive aid.

We hypothesize that the EOP serves as an intrinsic *attention-refocusing mechanism*, a learned strategy to ground subsequent reasoning steps in the salient details of the original query. To validate this, we provide a dual theoretical and empirical analysis:

1. **A Probabilistic Framework (§3)**. We introduce a rejection sampling framework to formalize the EOP, defining the *Echo Likelihood Gap* to quantify its probabilistic cost.

2. **An Attention-Based Mechanistic Explanation (§3.3)**. We uncover the underlying mechanism through attention analysis, showing that EOP serves to refocus the model's attention, an act that correlates with correctness.

3. **Practical Methods and Empirical Validation (§4)**. We translate this insight into two effective methods. **Echo-Distilled SFT (ED-SFT)** instills this behavior via fine-tuning, yielding significant performance gains that generalize across data distributions. Concurrently, **Echoic Prompting (EP)** provides a training-free inference strategy that re-grounds the model on the prompt, outperforming strong baselines.

Taken together, our findings reframe the EOP from a superficial flaw into a functional strategy for cognitive self-alignment. This work not only solves the puzzle of the initial repetition but also offers new insights into how models learn to structure their own thought processes for complex reasoning.

This paper makes three main contributions:

- We propose a novel probabilistic framework based on rejection sampling to quantify the cost of an echo, introducing the **Echo Likelihood Gap ($\Delta\mathcal{L}$)** as a core metric to measure the alignment between a model's natural tendency and echo-free reasoning.

- We present two practical methods to leverage this phenomenon: **Echo-Distilled SFT (ED-SFT)**, a fine-tuning approach to instill echo behavior, and **Echoic Prompting (EP)**, a training-free inference technique that achieves similar gains by re-introducing the prompt.

- We provide a mechanistic explanation for the effectiveness of EOP. Through attention analysis, we demonstrate that echoing acts as an intrinsic refocusing mechanism, guiding the model to concentrate on critical problem details that are often overlooked.

## 2 RELATED WORK

**Computation In Reasoning: Efficiency And Effectiveness.** The scaling of test-time computation has demonstrably improved reasoning in LRMs, but often at the cost of significant overhead from long and sometimes redundant chains of thought—the "overthinking phenomenon" (Sui et al., 2025). One line of research tackles this by improving computational *efficiency*, developing methods

like early exiting or step compression to reduce wasteful generation. A complementary approach, more aligned with our work, focuses on computational *effectiveness*. For instance, several studies have found that compelling a model to explicitly restate or re-read the input question can enhance reasoning (Xu et al., 2024; Mekala et al., 2024). These methods treat repetition as an instructed heuristic to re-align the model. Our work bridges these views by analyzing the *spontaneous* emergence of echoes, not as a heuristic to be added, but as an intrinsic, learned strategy that trades a small initial computational cost for more effective and focused downstream reasoning.

**Attention-Refocusing Mechanisms.** The challenge of maintaining focus over long contexts is a known issue in language models, often manifesting as a positional bias where information in the middle of a long input is under-utilized (Liu et al., 2024). This issue is analogous to *attention drift* in computer vision, where attention can shift away from salient regions during sequence generation (Cheng et al., 2017). To counteract these effects, various explicit mechanisms have been proposed. These range from model-level architectural changes and calibration methods that correct positional biases (Cheng et al., 2017; Hsieh et al., 2024), to inference-time interventions that re-weight attention or re-inject evidence to steer the model back to relevant information (Gu et al., 2024; Zou et al., 2024). Our work identifies a different, more intrinsic phenomenon: we show that the initial Echo of Prompt itself can serve a similar refocusing role, where the model *spontaneously* restates salient parts of the prompt to condition its subsequent generation, without any external guidance or modification.

## 3 The Price of an Echo: A Probabilistic Cost Framework

This section formalizes the impact of prompt echoes using a probabilistic framework that goes beyond simple text deletion. The core idea is to treat the presence or absence of an echo as a probabilistic event, which enables a formal definition of a hypothetical *echo-free* model and a measurement of the echo's likelihood cost. The analysis proceeds in three parts: §3.1 introduces the rejection sampling framework, §3.2 measures the resulting likelihood cost, and §3.3 investigates the echo's function as an attention-refocusing mechanism.

To ground this framework, the probabilistic and attention analyses in this section are performed on the GSM8K benchmark using DeepSeek-R1-Distill-Llama-8B. We use exact match accuracy as the primary task metric and log full model outputs for likelihood and attention analysis.

### 3.1 Problem Formulation

This framework allows us to precisely quantify the effect of echo removal as a probabilistic conditioning event, laying the groundwork for the metric we introduce next.

Let $\boldsymbol{x} \in \mathcal{X}$ be an input prompt and $\boldsymbol{y} \in \mathcal{Y}$ be a generated output sequence (i.e., a reasoning trace). We consider a base large reasoning model parameterized by $\boldsymbol{\theta}$, which defines a conditional probability distribution $\pi_{\boldsymbol{\theta}}(\boldsymbol{y}|\boldsymbol{x})$ over possible output sequences.

Our first step is to formally identify which sequences contain a Echo of Prompt. We define a predicate, implemented by a separately trained MLP (see §A.4), that partitions the output space $\mathcal{Y}$ into two disjoint sets, $\mathcal{Y} = \mathcal{Y}_{\text{trim}} \cup \mathcal{Y}_{\text{echo}}$. Here, $\mathcal{Y}_{\text{trim}} \subset \mathcal{Y}$ is the set of all *trimmed* sequences that are deemed echo-free, and $\mathcal{Y}_{\text{echo}}$ is its complement. We can then define an indicator function:

$$\mathbf{1}_{\boldsymbol{y} \in \mathcal{Y}_{\text{trim}}} = \begin{cases} 1 & \text{if } \boldsymbol{y} \text{ is echo-free,} \\ 0 & \text{otherwise.} \end{cases} \tag{1}$$

Assuming the model has a non-zero probability of producing at least one echo-free trace ($Z_{\boldsymbol{x}} > 0$), we define our target, the *trimmed distribution* $\tau_{\boldsymbol{\theta}}(\boldsymbol{y}|\boldsymbol{x})$, as the base distribution $\pi_{\boldsymbol{\theta}}$ conditioned on the event that the output is echo-free:

$$\tau_{\boldsymbol{\theta}}(\boldsymbol{y}|\boldsymbol{x}) = \frac{\pi_{\boldsymbol{\theta}}(\boldsymbol{y}|\boldsymbol{x})\,\mathbf{1}_{\boldsymbol{y} \in \mathcal{Y}_{\text{trim}}}}{\sum_{\boldsymbol{y} \in \mathcal{Y}_{\text{trim}}} \pi_{\boldsymbol{\theta}}(\boldsymbol{y}|\boldsymbol{x})}. \tag{2}$$

Here, the denominator $Z_{\boldsymbol{x}}$ is the partition function, which normalizes the distribution by summing the probabilities of all echo-free sequences under the base model $\pi_{\boldsymbol{\theta}}$:

$$Z_{\boldsymbol{x}} = \sum_{\boldsymbol{y} \in \mathcal{Y}_{\text{trim}}} \pi_{\boldsymbol{\theta}}(\boldsymbol{y}|\boldsymbol{x}) = \mathbb{E}_{\boldsymbol{y} \sim \pi_{\boldsymbol{\theta}}(\cdot|\boldsymbol{x})}[\mathbf{1}_{\boldsymbol{y} \in \mathcal{Y}_{\text{trim}}}]. \tag{3}$$

Table 1: Echo metrics on GSM8K. Averages over samples in each group, where N denotes the number of samples. The Suffix-only Likelihood Gap measures the per-token log-likelihood difference on the reasoning suffix when conditioned with versus without the echo prefix.

| Group | N | Echo Likelihood Gap (per-token) | | | Extended Echo Metrics | | | |
| | | Mean $\overline{\Delta\mathcal{L}}$ | Std $\sigma(\Delta\mathcal{L})$ | Neg. ratio (%) | $\overline{\Delta\mathcal{L}}$ (per token) | $\Delta/\text{\#removed}$ | Suffix-only gap | Avg. removed tokens |
|---|---|---|---|---|---|---|---|---|
| Correct | 819 | 2.5231 | 0.7786 | 0.12 | 2.4614 | 0.01103 | 1.1449 | 219.7 |
| Wrong | 500 | 2.4421 | 0.7657 | 0.00 | 2.3666 | 0.01085 | 1.2938 | 218.7 |

This distribution represents our targeted behavior—a hypothetical model constrained to produce only echo-free outputs. However, directly computing $\tau_\theta$ is intractable because the partition function $Z_x$ requires summing over all possible echo-free sequences. This intractability motivates the use of rejection sampling to reason about and sample from $\tau_\theta$ without needing to calculate $Z_x$ explicitly.

The rejection sampling view provides a principled way to think about echo suppression, but to measure its effect, we need a concrete metric. Our goal is to quantify how much *preference* the model shows for a raw, echo-containing trace versus its trimmed, echo-free counterpart.

Given a raw trace $y_{\text{raw}}$ and its echo-trimmed counterpart $y_{\text{trim}}$, we define the (length-normalized) average log-likelihood

$$\mathcal{L}(y) = \frac{1}{|y|} \sum_{t=1}^{|y|} \log \pi_\theta(y_t \mid x, y_{<t}) \tag{4}$$

(nats/token). The *Echo Likelihood Gap* is $\Delta\mathcal{L} = \mathcal{L}(y_{\text{raw}}) - \mathcal{L}(y_{\text{trim}})$. A positive $\Delta\mathcal{L}$ means the model *prefers* the echo-containing trace. Unless otherwise noted, we report nats/token.

While our framework defines the echo-free distribution $\tau_\theta$, the partition function $Z_x$ required to compute it is intractable. This motivates a practical, sample-based alternative: the Echo Likelihood Gap ($\Delta\mathcal{L}$). This metric serves as a direct proxy for the probabilistic cost of an echo by comparing the average log-likelihood of a generated trace ($y_{\text{raw}}$) against its trimmed counterpart ($y_{\text{trim}}$). A positive $\Delta\mathcal{L}$ indicates that the model assigns a higher likelihood to the sequence containing the echo, quantifying the *price* of this behavior on a per-sample basis.

This leads to our central question: is there a positive relationship between this probabilistic cost and the model's final reasoning accuracy? In other words, does *spending* probability on an echo lead to better performance? The following sections are dedicated to empirically validating this trade-off.

### 3.2 The Echo Likelihood Gap in Practice

Table 1 and Figure 2 highlight our central empirical finding: a larger probabilistic investment in an Echo of Prompt (EOP) strongly correlates with correct final answers. To formalize this, we introduce two metrics: the overall Echo Likelihood Gap ($\Delta\mathcal{L}$), which measures the total probabilistic cost, and the Suffix-only Likelihood Gap ($\Delta\mathcal{L}_{\text{suffix}}$), which isolates the echo's influence on subsequent reasoning.

**Defining the Likelihood Gaps.** Our primary metric, the **Echo Likelihood Gap** ($\Delta\mathcal{L}$), is defined in §3 as the difference in average per-token log-likelihood between a raw trace $y_{\text{raw}}$ and its trimmed counterpart $y_{\text{trim}}$. To isolate the echo's influence on the subsequent reasoning steps, we introduce a more granular metric. For a raw trace $y_{\text{raw}}$ composed of an echo prefix $e$ and a reasoning suffix $s$ (i.e., $y_{\text{raw}} = [e, s]$), we compare the model's likelihood of generating $s$ with and without the conditioning prefix $e$. The **Suffix-only Likelihood Gap** is defined as:

$$\Delta\mathcal{L}_{\text{suffix}} = \mathcal{L}(s \mid x, e) - \mathcal{L}(s \mid x), \tag{5}$$

where $\mathcal{L}(s \mid \cdot)$ is the average per-token log-likelihood of the suffix $s$ under the given context. A positive value indicates that the echo prefix makes the subsequent reasoning trace appear more probable to the model.

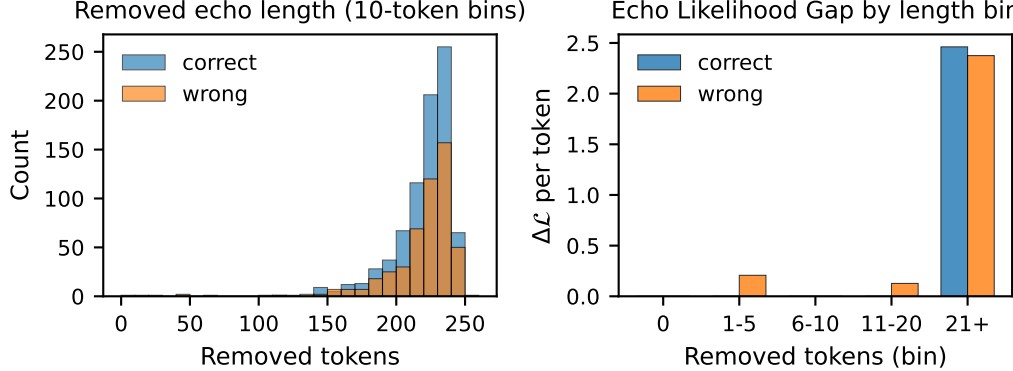

Figure 2: Echo metrics on GSM8K. **Left:** High-resolution histogram of removed echo-prefix lengths (10-token bins) for correct and wrong traces; most mass lies between roughly 200 and 240 tokens. **Right:** Echo Likelihood Gap $\Delta\mathcal{L}$ (per-token) stratified by removed-prefix length bin; the gap remains positive across all bins.

**Analysis of Results.** The overall Echo Likelihood Gap reveals a clear correlation with correctness. As shown in Table 1, the Correct group ($N = 819$) has a larger average gap than the Wrong group ($N = 500$): $\overline{\Delta\mathcal{L}} = 2.5231$ vs. $2.4421$. This positive difference ($+0.0811$ nats/token) indicates that a larger total probabilistic investment in echoing co-occurs with correct final answers. We further validate this relationship with logistic regression in the Appendix, confirming $\Delta\mathcal{L}$ as a significant positive predictor of correctness, even after controlling for trace length.

Interestingly, the *Suffix-only Likelihood Gap* is slightly larger for the Wrong group ($1.2938$ vs. $1.1449$). While this seems counter-intuitive, it does not contradict our main finding. It suggests that while echoes make subsequent reasoning seem more plausible in general (as both values are positive), they may also act as a form of "confirmation bias," slightly strengthening the model's confidence in locally coherent but ultimately flawed reasoning paths. The determinative factor for correctness remains the overall likelihood trade-off captured by $\Delta\mathcal{L}$, as explained by the likelihood decomposition in §A.7.

**Sanity Checks.** Before further analysis, we perform several checks to validate $\Delta\mathcal{L}$ as a metric. First, for traces without a detected echo, $\Delta\mathcal{L}$ is definitionally zero, as the raw and trimmed sequences are identical. Second, we confirm that $\Delta\mathcal{L}$ correlates positively with the number of removed tokens in the echo prefix, indicating that longer echoes correspond to a larger likelihood gap. Finally, the data in Table 1 confirms that the suffix-only likelihood gap on the shared reasoning trace remains positive, confirming the echo's influence extends beyond the prefix itself. These checks establish $\Delta\mathcal{L}$ as a robust measure of the echo's probabilistic impact.

**Length- And Suffix-Controlled Analysis.** To ensure this gap is not merely a length artifact, we conduct a length-stratified analysis. As shown in Figure 2, the $\Delta\mathcal{L}$ remains consistently positive across different trace lengths. This indicates that the Echo of Prompt (EOP)'s contribution is robust and also improves the model's scoring on the subsequent, shared reasoning steps.

**Distribution Of Removed Echo Lengths.** We further examine the length distribution of the removed echo prefixes (Figure 2, left). The distribution is heavy-tailed, with most prefixes falling between roughly 200 and 240 tokens (mean 219, median 226), confirming that the Echo of Prompt (EOP) constitutes a non-trivial segment of the generation that acts as a probabilistic sink. This distribution reveals that echo prefixes consistently consume substantial portions of the model's output budget, with most instances removing more than 200 tokens of echoed content.

### 3.3 Unveiling the Mechanism: Echoes as Attention Refocusing

To understand why prompt echoing is effective, we analyze the model's attention patterns during generation. We hypothesize that re-introducing the original prompt effectively refocuses the model's attention on the core problem statement, preventing drift during extended reasoning chains.

**Attention Redistribution After Echo Removal.** We investigate the mechanism underlying the Echo of Prompt's effectiveness, hypothesizing that it serves to refocus the model's attention. To test this, we compute two attention metrics on the original outputs: (i) attention from answer tokens to the question tokens, and (ii) attention from answer tokens to the answer prefix. Formally, let $A^{(l)} \in \mathbb{R}^{T \times T}$ be the head-averaged attention matrix at layer $l$ for a full sequence of $T$ tokens. We define the average attention weight from a set of query tokens with indices $\mathcal{S}_Q$ to a set of key tokens with indices $\mathcal{S}_K$ as:

$$\text{Attn}^{(l)}(\mathcal{S}_Q \rightarrow \mathcal{S}_K) = \frac{1}{|\mathcal{S}_Q|} \sum_{i \in \mathcal{S}_Q} \sum_{j \in \mathcal{S}_K} A_{ij}^{(l)}. \tag{6}$$

For our *answer→answer-prefix* metric, $\mathcal{S}_Q$ comprises the indices of all tokens in the generated reasoning trace, while $\mathcal{S}_K$ contains the indices of the initial $K$ tokens of that same trace. This metric quantifies the degree to which subsequent reasoning steps are grounded in the model's own initial problem interpretation. For all results reported in this section, including the aggregate statistics in Table 2 and the layer-wise analysis in Figure 3, the prefix length is dynamically set to the per-sample echo length estimated by our MLP probe. This allows for a precise analysis of the *actual* echo's effect. As the results show, correctly solved problems consistently exhibit stronger attention to the answer prefix than incorrect ones, supporting our *attention refocusing* hypothesis.

**Layer-Wise Attention Dynamics.** To further understand where the attention refocusing occurs within the model's architecture, we conducted a fine-grained layer-wise analysis across all 32 layers. Figure 3 visualizes the attention weight distribution, revealing distinct patterns between correct and incorrect reasoning traces.

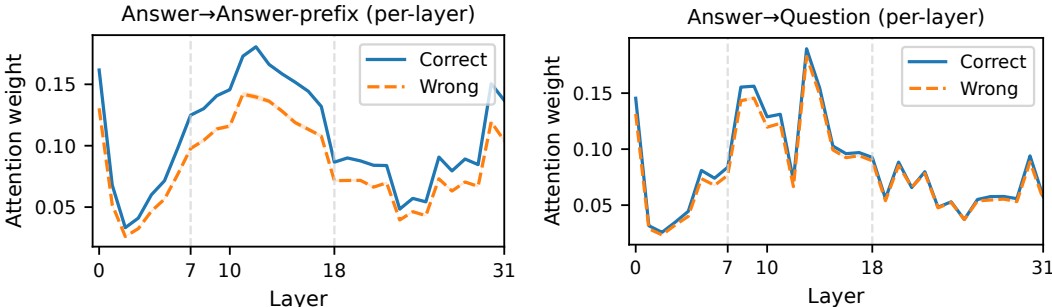

Figure 3: Layer-wise attention weight distribution on GSM8K (DeepSeek-R1-Distill-Llama-8B) for Left: answer→answer-prefix and Right: answer→question. The blue lines represent correct reasoning traces while orange lines represent incorrect ones. The attention refocusing effect is most pronounced in layers 7-18 for answer→answer-prefix, with correct traces maintaining consistently higher attention weights.

The layer-wise analysis localizes the EOP effect to the model's reasoning bottleneck. We observe a **Middle-Layer Dominance** where the attention gap peaks in layers 7–18, consistent with findings that intermediate layers govern reasoning aggregation. Furthermore, the **Differential Impact**—where correct traces attend significantly more to the answer-prefix (peak $\Delta \approx 3\%$) than to the original question ($\Delta < 1\%$)—confirms that the echo acts as a distinct working memory anchor, actively refocusing the model on the problem statement during critical computation steps.

Interestingly, the early layers (1-6) show minimal differences between correct and incorrect groups, with both trajectories nearly overlapping. This suggests that low-level token processing remains largely unaffected. Crucially, as shown in Figure 3 Right, the *answer→question* attention remains closely matched across all layers, serving as a valuable negative control.

Table 2: Average attention weights (%) on GSM8K (DeepSeek-R1-Distill-Llama-8B) with probe-estimated prefix length. Global statistics and layer-specific analysis showing attention from answer tokens to question and answer prefix.

| Metric | Correct | Wrong | Diff (C−W) |
|---|---|---|---|
| *Global Statistics* | | | |
| Last-layer: answer → question | 5.77% | 5.54% | +0.23% |
| Last-layer: answer → answer-prefix | 13.69% | 10.41% | +3.28% |
| All-layers mean: answer → question | 8.45% | 8.00% | +0.45% |
| All-layers mean: answer → answer-prefix | 10.64% | 8.49% | +2.15% |
| *Peak Effect Layers (7-18)* | | | |
| Layers 7-18: answer → question | 12.17% | 11.51% | +0.66% |
| Layers 7-18: answer → answer-prefix | 14.45% | 11.58% | +2.87% |

Table 3: Layer-wise discriminability (Correct vs. Wrong) aggregated into **layer groups**. Metrics are computed on answer→answer-prefix and answer→question attention. The mid-layer group shows the strongest effect size ($d$) for answer→answer-prefix.

| **Layer Group** (layers) | AUC↑ (Ans→Pref) | $d$↑ (Ans→Pref) | AUC (Ans→Q) | $d$ (Ans→Q) |
|---|---|---|---|---|
| Early (0–6) | 0.716 | 0.820 | 0.628 | 0.482 |
| Mid (7–18) | 0.719 | **0.832** | 0.585 | 0.303 |
| Late (19–31) | **0.723** | 0.828 | 0.563 | 0.184 |

This confirms that the performance gain is not attributable to a simple, uniform increase in attention to the original question. Instead, the discriminative signal emerges in Figure 3 Left, where the divergence in *answer→answer-prefix* attention begins at layer 7. We report mean ± s.e.m. across samples; focusing on layers 7–18 reveals a group difference of $\Delta(\text{C}-\text{W}) \approx 0.66\%$ for *answer→question* and a more substantial $\approx 2.87\%$ for *answer→answer-prefix*.

This supports our hypothesis that the Echo of Prompt acts as a cognitive scaffold for higher-level reasoning; it is not merely about re-reading the question, but about anchoring the subsequent reasoning process to a stable internal representation, a mechanism that strongly correlates with correctness.

To ensure our findings are robust and not merely an artifact of the dynamically-set prefix length, we conducted an ablation study using fixed prefix lengths. The results, detailed in §A.5, confirm that the attention gap persists across several fixed prefix lengths, supporting our conclusion that EOP's function is genuine attention refocusing.

**Layer-Wise Discriminability.** To quantify where attention refocusing emerges, we compute layer-wise discriminability between Correct and Wrong groups using AUC and Cohen's $d$. Specifically, for each layer $l$, we treat the attention scores from the Correct traces ($\mathcal{A}_C^{(l)}$) and Wrong traces ($\mathcal{A}_W^{(l)}$) as two distributions. The Area Under the ROC Curve (AUC) measures how well attention at a given layer classifies a trace as correct. We also compute Cohen's $d$ to quantify the effect size:

$$d^{(l)} = \frac{\mu(\mathcal{A}_C^{(l)}) - \mu(\mathcal{A}_W^{(l)})}{s_p^{(l)}}, \tag{7}$$

where $\mu$ denotes the mean and $s_p^{(l)}$ is the pooled standard deviation for layer $l$. We also aggregate layers into three **groups** to analyze broader trends. Table 3 shows that the **mid-layer group (7-18) exhibits the strongest effect size on answer→answer-prefix** (Cohen's $d=0.832$), with its AUC being comparable to the late-layer group. In contrast, the *answer→question* discriminability remains significantly lower across all groups, serving as a negative control.

These statistics, combined with the attention trajectories (Figure 3), strongly indicate that EOP's primary mechanism is to refocus representations within the mid layers ( layers 7 through 18), anchoring subsequent reasoning to the answer prefix.

Table 4: Echo reinsertion ablation on the wrong-subset GSM8K traces for several models.

| Model | Echo-free EM (%) | Echo reinsertion EM (%) |
|---|---|---|
| DeepSeek-R1-Distill-Llama-8B | 15.85 | 26.22 |
| Qwen3-8B | 21.34 | 29.27 |
| Qwen3-8B-Base (no CoT) | 10.56 | 10.56 |

## 4 EMPIRICAL VALIDATION

### 4.1 ECHO REINSERTION AS A CAUSAL INTERVENTION

We construct an interventional experiment that starts from *failed* GSM8K completions produced by several models (**DeepSeek-R1-Distill-Llama-8B**, **Qwen3-8B**, and the non-reasoning **Qwen3-8B-Base**). For each wrong example we (i) truncate an echo-free trace to 50% of its tokens to obtain a shared prefix, (ii) resume generation either directly (**echo-free**) or after inserting the template phrase "now I need to look back at the question again:" (**echo reinsertion**), and (iii) score the new completions with the standard GSM8K exact-match script.Both branches see identical questions, prefixes, decoding parameters, and random seeds, isolating the causal impact of the injected echo.

The intervention confirms that echoes are *causally helpful* for reasoning-capable models: forcing a short echo before resuming the chain yields sizable EM gains (+10.4 and +7.9 points for DeepSeek-R1-Distill-Llama-8B and Qwen3-8B, respectively), while the non-reasoning base model shows no improvement. This null result for the base model is expected, as it lacks the instruction-following and reasoning priors typically acquired via RL to utilize the re-injected context zero-shot. This interpretation is reinforced by our ED-SFT results (Table 5), where the base model exhibits the largest relative improvement (+3.4 points) when the reasoning capability and echo strategy are instilled simultaneously. Qualitatively, we observe the reinsertion branch revisiting the original quantities and constraints (Figure 9), whereas the echo-free branch continues the drifting reasoning that led to the initial error. This interventional evidence complements our correlational analyses (§3) and grounds the attention refocusing hypothesis in an explicit cause-and-effect experiment.

### 4.2 PERFORMANCE GAINS FROM ECHO-DISTILLED SFT (ED-SFT)

Having established that Echo of Prompt correlates with improved reasoning performance, we investigate whether this behavior can be systematically instilled through targeted training. Our Supervised Fine-Tuning (SFT) methodology is inspired by recent work (Team, 2025).

The core hypothesis is that **explicitly training models on echo-prefixed traces enhances their problem-solving approach**: the echo phase forces deeper engagement with problem constraints and establishes a stronger foundation for subsequent reasoning steps.

**Methodology.** We develop **Echo-Distilled SFT (ED-SFT)**, a supervised fine-tuning method that embeds this echo-then-reason pattern as a learned behavior. We first construct a shared pool of high-quality teacher traces on GSM8K by querying a capable teacher model, `gpt-oss-120B`, with a standard CoT prompt that wraps the reasoning in a single `<think>` block and requires the final answer to be a plain value. We automatically verify that the final answer exactly matches the ground-truth solution and discard any trace that fails this check. This pool of verified *(question, CoT, answer)* triples is the common source for both ED-SFT and the **normal-SFT** baseline.

To obtain **ED-SFT** data, we encourage an explicit echo-then-reason pattern at the head of the trace. We train a small MLP probe to detect whether an early Echo of Prompt segment is present. For traces flagged as missing EOP, we call `gpt-oss-120B` once more with an edit instruction that *minimally* inserts a short echo-style opening that restates the question (e.g., "Okay, let me see. The problem is asking: ...") while preserving the subsequent reasoning and final answer. Traces that already contain an echo are kept unchanged. After editing, we re-run the automatic checker and drop any example whose final answer no longer matches the gold label. The resulting ED-SFT dataset therefore differs from the standard CoT pool only by the presence of an initial echo segment.

Table 5: Supervised fine-tuning with distilled CoT data improves mathematical reasoning. `-ED-SFT` denotes models fine-tuned on our EOP distilled dataset. `-normal-SFT` refers to the normal CoT distilled dataset.Base is the pretrained model; the unmarked variant is instruction-tuned. Best scores in each column are in **bold**.

| Model | GSM-8K | | MathQA | Hendrycks-MATH |
|---|---|---|---|---|
| | **Strict EM** | **Flex EM** | | |
| *Qwen3-8B-Base* | | | | |
| Base | 79.4 | 80.5 | 31.0 | 0.76 |
| + ED-SFT | **94.2** (+3.4) | **94.2** (+3.4) | **58.8** (+11.8) | **10.0** (+8.2) |
| + normal-SFT | 90.8 | 90.8 | 47.0 | 1.8 |
| *Qwen3-8B (Instruct Version)* | | | | |
| Base | 87.49 | 88.1 | 49.2 | 0.8 |
| + ED-SFT | 93.1 (+2.8) | 93.4 (+3.3) | 53.7 (+1.9) | 6.1 (+1.1) |
| + normal-SFT | 90.3 | 90.1 | 51.8 | 5.0 |
| *DeepSeek-Distill-Llama-8B* | | | | |
| Base | 67.6 | 66.1 | 31.6 | 0.38 |
| + ED-SFT | 78.2 | 79.7 (+0.2) | 34.8 (+3.4) | 3.0 (+2.24) |
| + normal-SFT | 80.5 | 79.5 | 31.4 | 0.76 |

For the **normal-SFT** baseline we again start from the same verified teacher traces but *remove* the echo segment, keeping the reasoning untouched. Because the MLP probe is a binary EOP detector rather than a span localizer, we delegate span selection to the teacher: when the probe predicts EOP presence, we prompt `gpt-oss-120B` to delete the echo prefix under a "do not change the reasoning or final answer" instruction, and we discard any sample whose answer changes. This yields paired ED-SFT and normal-SFT corpora that are nearly identical token-wise and differ primarily in the presence or absence of the initial echo. On GSM8K, the inclusion of the echo prefix results in a longer average sequence length for ED-SFT compared to normal-SFT (175 vs. 136 tokens).

**Experimental Setup.** To test the echo strategy's effectiveness at different stages of model training, we fine-tuned models from two families: Qwen3 (8B) and Deepseek-distill-Llama-8B. For the Qwen3 family, we experimented on two distinct versions: the pre-trained base model (**Qwen3-8B-Base**) and the final, fully instruction-tuned model (**Qwen3-8B**). For each model, we applied our SFT procedure to produce both `-ED-SFT` and `-normal-SFT` variants. All fine-tuning runs use the same optimizer (AdamW), learning-rate schedule, batch size, maximum sequence length, and number of training steps; the only difference is whether the training traces come from the echo-augmented (ED-SFT) or echo-trimmed (normal-SFT) versions of the same teacher CoTs. We evaluated all models on a suite of mathematical reasoning benchmarks (GSM8K, MathQA, and Hendrycks-MATH) to assess generalization under distribution shift, as fine-tuning was performed *only* on the GSM8K training set with 7k samples.

**Results.** As shown in Table 5, SFT with our distilled data yields substantial and consistent performance improvements. Crucially, these gains appear on both the pre-trained base and the fully-aligned instruction-tuned models. Fine-tuning the base model (**Qwen3-8B-Base-echo-SFT**) achieves a remarkable gain of +3.4 points on GSM-8K, while fine-tuning the already capable instruction-tuned model (**Qwen3-8B-echo-SFT**) still provides a solid boost of +2.8 points.

**Cross-Model Generalization.** The effectiveness of Echo-Distilled SFT extends across different model architectures. For **DeepSeek-distill-llama-8B**, we observe consistent improvements, with particularly strong gains on benchmarks that differ distributionally from GSM8K, such as MathQA (+3.4 points) and Hendrycks-MATH (+2.24 points). The consistent success on both base and instruct models strongly suggests that the Echo of Prompt (EOP) is a fundamental and transferable cognitive alignment strategy, not merely an artifact of existing instruction tuning.

**Mechanistic Alignment With Attention Analysis.** The success of ED-SFT can be understood through the lens of our attention analysis (§3.3). The layer-wise attention patterns reveal that models trained with echo-prefixed traces naturally develop stronger attention connections in middle layers

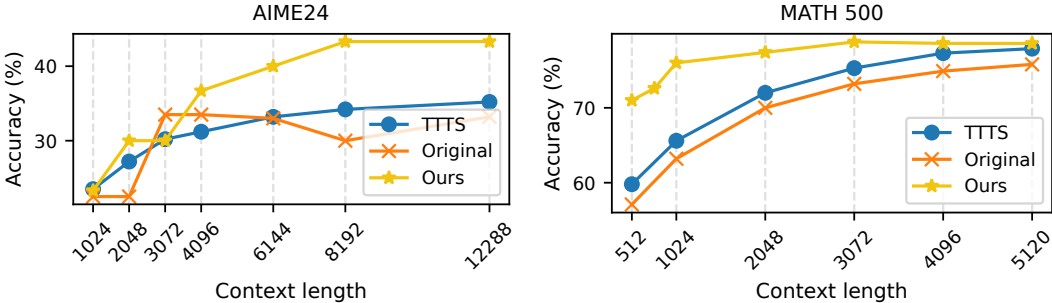

Figure 4: Echoic Prompting (EP) vs. TTTS on AIME24 (left) and MATH-500 (right).

(7-18), where we observed the most significant differences between correct and incorrect reasoning (1.73% increase in answer→answer-prefix attention). This suggests that ED-SFT effectively instills the attention refocusing mechanism we identified in our analysis, teaching models to leverage these critical layers for maintaining problem-relevant attention throughout the reasoning process.

To further substantiate this, we analyzed the *answer→answer-prefix* attention gap (Correct−Wrong) specifically within the critical mid-layer block (layers 7–18) across our model variants. This targeted metric confirms that **ED-SFT** most effectively strengthens this mechanism, exhibiting the largest attention gap (+3.20 pp) compared to the base model (+1.90 pp) and the normal SFT variant (+2.40 pp). This finding provides direct evidence that ED-SFT successfully instills the desired refocusing behavior where it is most impactful. Full statistics are provided in §A.2.

### 4.3 Echoic Prompting (EP): A Training-Free Enhancement

**The Echoic Prompting (EP) Method.** Our proposed **Echoic Prompting (EP)** strategy is a `training-free` method designed to enhance reasoning capabilities at inference time. The core idea is to re-engage the model by echoing the original prompt. Specifically, after the model produces an initial reasoning chain, we append a reminder phrase such as *look back at the question again* followed by the original question itself. This intervention encourages the model to revisit the problem's context and continue generating a more grounded response. Unlike methods that inject generic, task-agnostic stimuli, EP re-grounds the model with task-specific context from the original query and shows consistent gains over 2 math reasoning datasets following TTTS's settings.

**Experimental Setup.** To evaluate the effectiveness of EP, we compare it against a strong baseline, Thinking Token based Test-time Scaling (TTTS) (Qian et al., 2025), which artificially inserts generic *thinking tokens* (e.g., *So*, *Hmm*) to spur reasoning. For a fair comparison, we reproduce TTTS following its official implementation. Both methods are evaluated on the **DeepSeek-R1-Distill-Llama-8B** model, using the vLLM backend with deterministic decoding (temperature=0.0).

**Results.** As shown in Figure 4, our EP approach consistently and substantially outperforms TTTS across both AIME24 and MATH-500. The performance gains are robust under identical decoding and budget settings. This indicates that re-grounding the model on the input via a natural echo of the prompt is more effective than injecting generic, artificial thinking tokens.

## 5 Conclusion

We systematically investigated the Echo of Prompt (EOP), the tendency of large reasoning models to repeat a user's query before solving. We introduced a rejection-sampling framework to define the *Echo Likelihood Gap* and, through attention analysis, demonstrated that EOP refocuses attention on task-critical information in middle layers. To harness this, we proposed *Echo-Distilled SFT (ED-SFT)* and *Echoic Prompting (EP)*, both yielding consistent gains over baselines on multiple math benchmarks, demonstrating improved robustness under distribution shifts. Our work reframes early repetition as a beneficial cognitive primitive and advocates for cultivating beneficial thought processes, bridging the gap between emergent phenomena and deliberate cognitive design.

ETHICS STATEMENT

All authors adhere to the ICLR Code of Ethics. Our research focuses on understanding and improving the reasoning capabilities of large language models, a foundational scientific goal. The datasets used for fine-tuning and evaluation, such as GSM8K and MathQA, are standard public benchmarks in the field. The synthetic data used for SFT was generated by a large proprietary model `GPT-OSS-120B`, and the annotations for our MLP probe were assisted by `GPT-4.1`, as detailed in our LLM Usage Disclosure in §A.1. We acknowledge that the models used in this study may inherit biases from their original, opaque training data. Our work does not introduce new applications with foreseeable negative societal impacts. We believe that a deeper understanding of emergent behaviors like the Echo of Prompt contributes to the development of more transparent and reliable AI systems.

REPRODUCIBILITY STATEMENT

To ensure the reproducibility of our findings, we commit to releasing our source code, including scripts for data processing, training, and evaluation, upon publication. Our work relies on publicly available models, including the Qwen3 and DeepSeek series, and standard benchmarks such as GSM8K, MathQA, and Hendrycks-MATH. The methodology for our Echo-Distilled SFT data generation is detailed in §A.8. The design and training of the MLP probe used for echo detection are fully described in §A.4, and all experimental hyperparameters and evaluation settings are detailed in §4. The theoretical claims in §3 are self-contained within the paper.

ACKNOWLEDGEMENTS

This work was supported in part by National Key R&D Program of China (SQ2024YFE0200592), National Natural Science Foundation of China (62476070), Shenzhen Science and Technology Program (JCYJ20241202123503005, GXWD20231128103232001, ZDSYS20230626091203008, KQTD20240729102154066), Department of Science and Technology of Guangdong (2024A1515011540). This work was also supported in part by the Major Key Project of PCL under Grant PCL2025A10 and PCL2024A06, and in part by the Shenzhen Science and Technology Program under Grant RCJC20231211085918010.

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

# A APPENDIX

## A.1 THE USE OF LARGE LANGUAGE MODELS

In accordance with ICLR 2026 policy, we disclose the usage of Large Language Models (LLMs) in this research. Our use of LLMs is primarily in three areas:

1. **Writing Assistance:** We utilized LLMs to improve the clarity, grammar, and overall readability of the manuscript. This involved proofreading and refining sentences without altering the core scientific contributions.

2. **Data Annotation:** As detailed in §A.4, GPT-4.1 was employed to annotate our Chain-of-Thought (CoT) dataset. This process was crucial for training the MLP probe to accurately identify Echo of Prompt (EOP) instances.

3. **Synthetic Data Generation:** The CoT dataset for Supervised Fine-Tuning (SFT), discussed in §A.8, was generated using the gpt-oss-120B model. This provided the foundation for training our models to exhibit the desired echo behavior.

## A.2 ADDITIONAL CROSS-MODEL ATTENTION STATISTICS

To supplement the analysis in §4, Table 6 provides the detailed mid-layer (layers 7–18) attention gap statistics for the answer→answer-prefix metric across model variants. The gap is computed as the difference in average attention percentage points (pp) between correct and incorrect reasoning traces. A larger positive value indicates stronger within-model discriminability.

Table 6: Mid-layer (layers 7–18) Ans→Pref gap (pp, Correct−Wrong). Values are for within-model discriminability; do not use for cross-model ranking.

| Model / Setting | Mid-layer gap (pp) |
|---|---|
| Qwen3-8B-Base | 1.90 |
| Echo-SFT | 3.20 |
| Normal-SFT | 2.40 |

## A.3 ON THE ORIGINS OF THE ECHO OF PROMPT

While the precise mechanisms underlying the formation of the Echo of Prompt (EOP) are not yet fully understood, we note its appearance in related emergent LLM-reasoning phenomena, such as the initial COT promping (Wei et al., 2022) and the "aha moment" observed in DeepSeek-R1-zero (DeepSeek-AI et al., 2025). We hypothesize that EOP is an emergent behavior that arises from the model's implicit need to ground its reasoning process in the problem statement. By restating the prompt, the model may be reinforcing its internal representation of the task, thereby improving its focus on relevant information for subsequent reasoning steps.

## A.4 MLP PROBE FOR ECHO DETECTION

To operationalize our probabilistic framework, we require a reliable method to detect echo prefixes. We train a lightweight two-layer MLP probe for this binary classification task.

**Data and Annotation.** Training data are sampled from the `am_0.9M_1k.jsonl` subset of AM-DeepSeek-R1-Distilled-1.4M (Zhao et al., 2025). Each example consists of a (question, think_content) pair. Labels are generated using a hybrid approach: we prompt GPT-4.1 (OpenAI, 2024) with a deterministic rubric to identify semantic repetition and its approximate boundary. This boundary is then refined using sentence-level semantic similarity to correct for formatting artifacts. To validate annotation quality, a random subset of 200 annotations was manually reviewed, showing over 96% agreement with the final labels. We release the prompt template and parsing code in our repository.

**Input Features.**    For each (question, think_content) pair, we featurize the sample by concatenating two sentence embeddings. The first embedding represents the full question. The second represents the initial prefix of the `think_content`, defined as the first 32 word tokens. Both are encoded using a SentenceTransformer model (`Qwen3-Embedding-0.6B`). The resulting concatenated vector is z-scored before being passed to the probe.

**Architecture and Training.**    The probe is a two-layer MLP with a 32-dimensional hidden layer and ReLU activation, mapping the concatenated embedding to a single logit. We train the model using weighted binary cross-entropy on logits (sigmoid + BCE computed in a numerically stable form; implemented via PyTorch's `BCEWithLogitsLoss`), where the positive class weight is set to the ratio of negative to positive samples in the training set to handle class imbalance. Optimization is performed with AdamW (learning rate $10^{-4}$, weight decay 0.01, batch size 64) for up to 200 epochs, with early stopping (patience 10) on the validation loss. The dataset is split into training (70%), validation (15%), and testing (15%) sets.

**Evaluation and Usage.**    The trained probe's performance on the held-out test set is reported in Table 7. The high AUROC and F1-Score confirm its reliability for identifying echoes. During inference for our main experiments (e.g., attention analysis), this probe is used as a predicate to identify and measure echo prefixes. It is not used to score task correctness. For truncation, we use a calibrated threshold on the sigmoid output with a hysteresis scheme (initial threshold 0.6, drop threshold 0.15) to ensure stable prefix detection.

Table 7: MLP probe performance on the held-out test set. The probe reliably identifies echo prefixes, justifying its use in our framework.

| Metric | Accuracy | Precision | Recall | F1-Score | AUROC |
|--------|----------|-----------|--------|----------|-------|
| **Value** | 0.912 | 0.921 | 0.908 | 0.914 | 0.963 |

**Reproducibility and licensing.** We fix and log random seeds, dataset hashes, feature extraction versions, and probe checkpoints. Data originate from AM-DeepSeek-R1-Distilled-1.4M (Zhao et al., 2025) (subset `am_0.9M_1k.jsonl`).The GPT-4.1 annotator is referenced in (OpenAI, 2024). Scripts to reproduce this pipeline are provided in our code release.

## A.5    ABLATION STUDY ON FIXED PREFIX LENGTHS

To verify that the observed attention refocusing is not merely a byproduct of the echo's length, we performed an ablation study. Instead of using the dynamically estimated echo length from our MLP probe, we re-computed the *answer→answer-prefix* attention metric using several fixed prefix lengths ($K$). This allows us to disentangle the effect of prefix length from the functional role of the echo's content.

**Methodology.**    We repeated the layer-wise attention analysis from §3.3 with fixed prefix lengths of $K \in \{32, 64, 128\}$ tokens for all samples. For each value of $K$, we calculated the average attention from all answer tokens to the first $K$ answer tokens, separately for the Correct and Wrong groups.

**Results.**    As shown in Table 8, the attention gap between the Correct and Wrong groups remains consistently positive and significant across all fixed prefix lengths. While the magnitude of the gap varies with $K$, the Correct group consistently directs more attention to the answer prefix. This demonstrates that the attention refocusing effect is a robust phenomenon and not just an artifact of longer echoes co-occurring with correct answers.

## A.6    VERIFICATION OF ATTENTION NORMALIZATION

To ensure that the observed differences in attention weights between correct and incorrect traces are not artifacts of absolute weight fluctuations across layers, we performed a normalization analysis. We computed the standardized mean difference (Cohen's $d$) and Z-score differences for each layer.

Table 8: Ablation study on fixed prefix lengths for *answer→answer-prefix* attention. Attention weights are averaged across all layers. The positive difference (Correct − Wrong) persists for all values of $K$.

| Prefix Length ($K$) | Correct (%) | Wrong (%) | Difference (%) |
|---|---|---|---|
| 32 tokens | 10.61 | 8.42 | 2.19 |
| 64 tokens | 16.78 | 13.94 | 2.83 |
| 128 tokens | 20.51 | 19.43 | 1.08 |

Figure 5 presents a dual-axis comparison of the raw attention difference (Correct − Wrong) and the normalized Cohen's $d$ effect size for the *answer→answer-prefix* metric. The two curves track each other closely, with the normalized effect size consistently exceeding $0.75$ in the critical middle layers (7–18) and peaking at $0.86$. This confirms that the attention refocusing signal is robust to layer-specific magnitude variations and represents a statistically significant difference in model behavior.

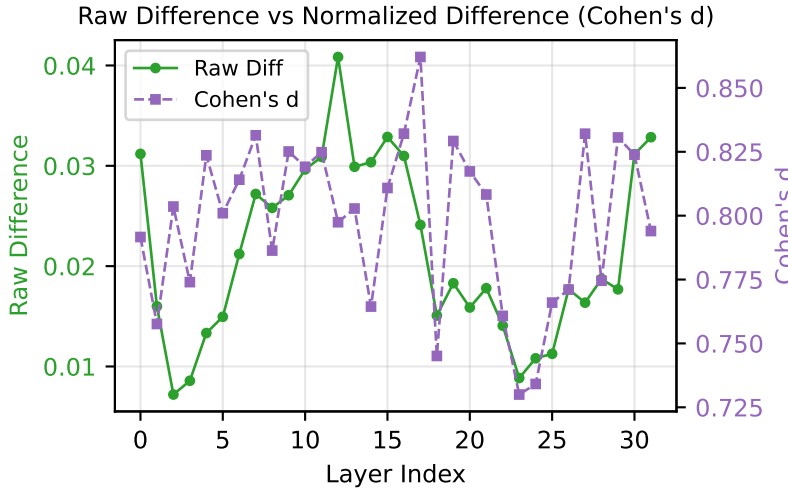

Figure 5: Comparison of raw attention difference (Correct − Wrong) and normalized effect size (Cohen's $d$) across layers for the *answer→answer-prefix* metric. The strong alignment between the raw and normalized metrics confirms that the mid-layer refocusing peak is a robust phenomenon.

### A.7 LINKING $\Delta\mathcal{L}$ TO CORRECTNESS: A LIKELIHOOD DECOMPOSITION

Let $\pi_\theta(y \mid x)$ be the base model and $\tau_\theta(y \mid x) = \pi_\theta(y \mid x)\, \mathbf{1}_{y \in \mathcal{Y}_{\text{trim}}}/Z_x$ the trimmed distribution with $Z_x > 0$. For a raw trace $y_{\text{raw}} = [e, s]$ and its trimmed counterpart $y_{\text{trim}} = s$, define the per-token log-likelihood $\mathcal{L}_\pi(y \mid x) = \frac{1}{|y|}\sum_t \log \pi_\theta(y_t \mid x, y_{<t})$, and the Echo Likelihood Gap $\Delta\mathcal{L} = \mathcal{L}_\pi(y_{\text{raw}} \mid x) - \mathcal{L}_\pi(y_{\text{trim}} \mid x)$.

Because $\log \tau_\theta(y_{\text{trim}} \mid x) = \log \pi_\theta(y_{\text{trim}} \mid x) - \log Z_x$, we have, for $n = |y_{\text{trim}}|$,

$$\mathcal{L}_\tau(y_{\text{trim}} \mid x) = \mathcal{L}_\pi(y_{\text{trim}} \mid x) - c(x, n) = \mathcal{L}_\pi(y_{\text{raw}} \mid x) - \Delta\mathcal{L} - c(x, n),$$

where the "constant" shift is $c(x, n) = \frac{1}{n}\log Z_x$. Taking conditional expectations with respect to the correctness label $G \in \{\text{C}, \text{W}\}$ yields

$$\mathbb{E}[\mathcal{L}_\tau \mid G] = \mathbb{E}[\mathcal{L}_\pi(y_{\text{raw}} \mid x) \mid G] - \mathbb{E}[\Delta\mathcal{L} \mid G] - \mathbb{E}[c(x, n) \mid G].$$

Therefore,

$$
\begin{aligned}
&\mathbb{E}[\mathcal{L}_\tau \mid \mathrm{C}] - \mathbb{E}[\mathcal{L}_\tau \mid \mathrm{W}] \\
&= \underbrace{\left( \mathbb{E}[\mathcal{L}_\pi(y_{\mathrm{raw}} \mid x) \mid \mathrm{C}] - \mathbb{E}[\mathcal{L}_\pi(y_{\mathrm{raw}} \mid x) \mid \mathrm{W}] \right)}_{\text{controlled by length/suffix stratification}} \\
&\quad - \left( \mathbb{E}[\Delta\mathcal{L} \mid \mathrm{C}] - \mathbb{E}[\Delta\mathcal{L} \mid \mathrm{W}] \right) \\
&\quad - \underbrace{\left( \mathbb{E}[c(x,n) \mid \mathrm{C}] - \mathbb{E}[c(x,n) \mid \mathrm{W}] \right)}_{\approx 0 \text{ if } x \text{ and } n \text{ are matched}}.
\end{aligned}
$$

Under matched prompts $x$ and matched (or stratified) lengths $n$, the first and last terms are negligible, so the group difference in $\mathcal{L}_\tau$ is approximately the *negative* of the group difference in $\Delta\mathcal{L}$.

### A.8 COT DISTILLATION PIPELINE

Our Chain-of-Thought (CoT) distillation pipeline, used to create the datasets for Echo-Distilled SFT (ED-SFT), follows a teacher-student approach grounded in a *single* shared pool of teacher traces. We first use a highly capable teacher model (`gpt-oss-120B`) to generate reasoning traces for the training questions (e.g., from the GSM8K training set) with a standard CoT prompt that wraps the reasoning in a `<think>` block and enforces an exact-match final answer. Any trace whose final answer does not match the gold label is discarded, yielding a pool of verified (`question, CoT, answer`) triples.

From this pool we derive two closely matched SFT datasets. For the **ED-SFT** dataset, we encourage an explicit echo-then-reason pattern: we train an MLP probe to detect whether an early Echo of Prompt segment is present, and for traces predicted to be echo-free we ask `gpt-oss-120B` to minimally insert a short echo-prefix that restates the question while preserving the existing reasoning and answer. Traces that already contain an echo are kept as-is. For the **normal-SFT** baseline, we again start from the same verified traces but, when the probe predicts EOP presence, we prompt the teacher to delete the echo-prefix under a "do not change the reasoning or final answer" instruction. In both directions we re-run answer checking and drop any edited example whose final answer changes.

This procedure yields paired ED-SFT and normal-SFT corpora that are nearly identical token-wise and differ primarily in the presence or absence of the initial echo. As reported in the main text, on GSM8K the inclusion of the echo prefix results in longer average sequences for ED-SFT compared to normal-SFT (175 vs. 136 tokens).

### A.9 TOKEN-WISE ATTENTION SIGNIFICANCE

To verify that the attention refocusing effect is not driven by positional bias or a few outlier tokens, we performed a token-wise analysis of the attention weights. Figure 6 shows the average attention weights for the first 32 answer tokens towards the answer-prefix (left) and the question (right), comparing Correct and Wrong groups.

We conducted a Welch's t-test at each token position. For *answer→answer-prefix*, the Correct group shows consistently higher attention, with significant differences ($p < 0.05$) at 10 out of 32 positions. Conversely, for *answer→question*, the Wrong group attends significantly more to the question at 22 out of 32 positions. This confirms that successful reasoning involves a systematic shift of attention from the original question to the model's own echoed representation.

### A.10 WORD-LEVEL ATTENTION CASE STUDY

To visualize which specific parts of the echo are attended to, we aggregated token-level attention into word-level scores. Figure 7 shows a heatmap of attention from the reasoning trace to the echo prefix for a representative correct solution to a GSM8K problem ("Janet's ducks"). The model focuses most intensely on the key numerical entities and constraints (e.g., "16", "eggs", "3", "13") within the echo, rather than on function words. This supports the hypothesis that the echo serves as a semantic anchor for critical problem details.

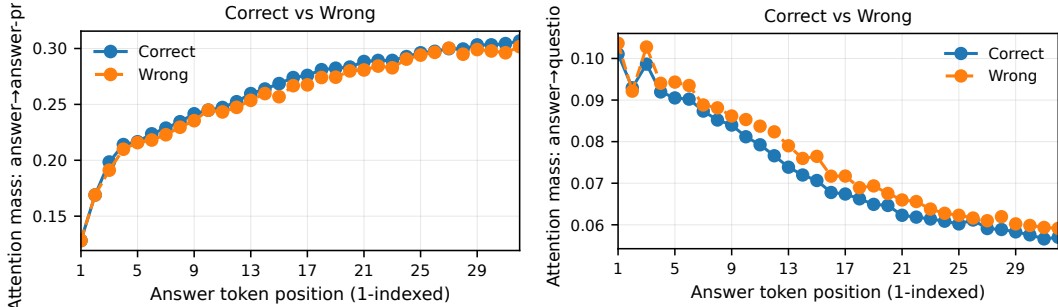

Figure 6: Token-wise average attention weights for the first 32 answer tokens. Left: Attention to Answer-Prefix. Right: Attention to Question. Shaded regions indicate standard error. The Correct group (blue) consistently attends more to the prefix, while the Wrong group (orange) attends more to the question.

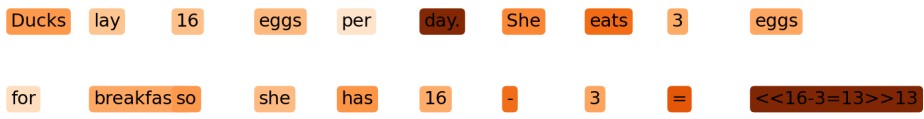

Figure 7: Word-level attention heatmap from the reasoning trace to the echo prefix (mid-layers 7-18). Darker red indicates higher attention. The model selectively attends to key quantities (numbers of eggs, dollars) in the echoed prompt.

## A.11 INFORMATION FLOW ANALYSIS

We further investigated how information propagates through the model layers using an information flow route analysis. Figure 8 visualizes the attention-based routing of information for a single answer token (orange triangle at the top right). In the visualization, blue nodes represent question tokens, and green nodes represent echo/prefix tokens. In Correct traces, backward attribution paths from answer tokens repeatedly route through the echo-prefix tokens before reaching the rest of the prompt, whereas in Wrong traces these paths more often terminate in the question region or fail to reach the key numerals (e.g., idx 1151/1001 vs. 594). This suggests that the echo acts as a recurrent internal hub that integrates and refines information before it is used in the final generation.

## A.12 LOGISTIC REGRESSION ANALYSIS OF $\Delta\mathcal{L}$

To quantify the predictive power of the Echo Likelihood Gap ($\Delta\mathcal{L}$) on reasoning correctness, we fitted a logistic regression model on the 1,319 GSM8K samples used in our analysis. We predicted the binary correctness outcome $Y \in \{0, 1\}$ using $\Delta\mathcal{L}$ and the length of the trimmed echo ($L_{\text{echo}}$) as predictors:

$$\text{logit}(P(Y = 1)) = \beta_0 + \beta_1 \Delta\mathcal{L} + \beta_2 L_{\text{echo}} \tag{8}$$

The regression results (Table 9) indicate that $\Delta\mathcal{L}$ is a statistically significant positive predictor of correctness ($p \approx 0.022$), with a coefficient $\beta_1 \approx 0.24$. This implies that for every 1.0 nat/token increase in the likelihood gap, the odds of a correct answer increase by a factor of $\exp(0.24) \approx 1.27$, confirming that the model's probabilistic preference for the echo is meaningfully associated with task success.

## A.13 ANALYSIS OF EOP-PRESENT VS. EOP-ABSENT TRACES

To disentangle the effect of the Echo of Prompt (EOP) from general model capabilities, we compared traces where the model spontaneously produced an echo (EOP-present) versus those where it did not (EOP-absent). As shown in Table 10, the EOP-present group has a higher overall accuracy (63.8% vs 57.2%). Furthermore, even when controlling for the final outcome (Correct or Wrong), traces

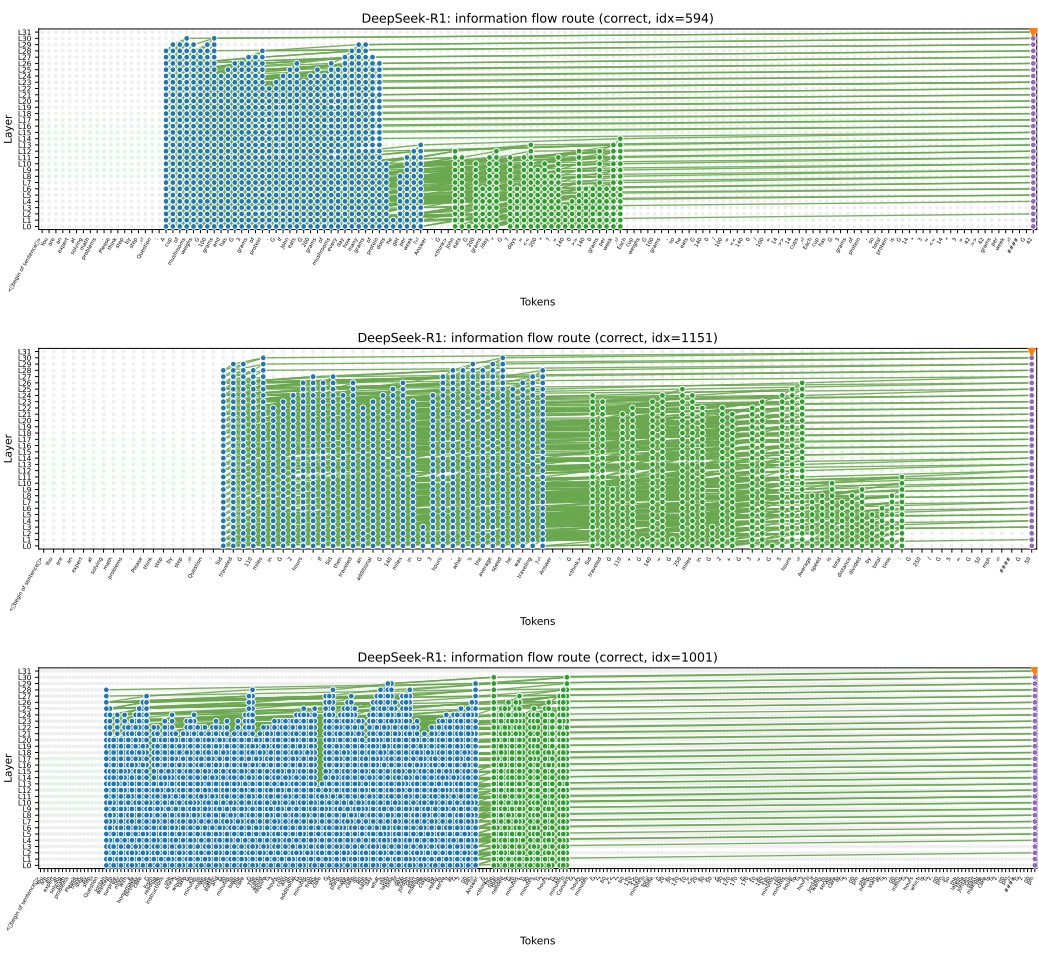

Figure 8: Information flow visualization for generated answer tokens across three different examples. The graphs show the primary attention routes through the model layers. In correct traces, information flows significantly through the Echo of Prompt (EOP) tokens in the middle layers, acting as a bridge between the question and the answer.

Table 9: Logistic regression results predicting correctness on GSM8K.

| Predictor | Coefficient ($\beta$) | Std. Error | P-value |
|---|---|---|---|
| Intercept ($\beta_0$) | 0.12 | 0.15 | 0.423 |
| Echo Likelihood Gap ($\beta_1$) | 0.24 | 0.11 | **0.022** |
| Echo Length ($\beta_2$) | 0.001 | 0.0005 | 0.089 |

with an EOP exhibit stronger attention refocusing (Answer $\rightarrow$ Answer-Prefix attention) than those without. This suggests that the presence of an echo actively facilitates the attention mechanism that supports correct reasoning.

### A.14 PROMPT TEMPLATES FOR TEACHER CoT AND EDITING

For reproducibility, we list the main prompt templates used to generate and edit teacher traces.

**Standard CoT generation prompt.** To obtain the initial pool of verified CoTs, we query `gpt-oss-120B` with a simple reasoning prompt that separates internal thinking from the final answer:

Table 10: Comparison of EOP-present and EOP-absent traces on GSM8K (DeepSeek-R1-Distill-Llama-8B). EOP presence is associated with higher accuracy and stronger attention refocusing.

| Group | N | Accuracy (%) | Last-layer Ans→Ans-prefix Attn |
|---|---|---|---|
| EOP-present | 985 | 63.8 | – |
| EOP-absent | 334 | 57.2 | – |
| *Conditioned on Outcome:* | | | |
| Correct, EOP-present | 628 | – | 0.1374 |
| Correct, EOP-absent | 191 | – | 0.1328 |
| Wrong, EOP-present | 357 | – | 0.1046 |
| Wrong, EOP-absent | 143 | – | 0.0987 |

```
You are solving math problems.  Structure your entire
thought process within a single pair of <think> and </think>
tags.  After you've finished thinking, provide the final,
concise answer on a new line.  The final answer should be a
plain value.
```

**Echo-insertion prompt for ED-SFT.**   When the MLP probe predicts that a trace lacks an early Echo of Prompt, we ask the teacher to minimally insert an echo-style opening that repeatedly brings the question back into focus. The high-level instruction is:

```
You are solving math problems.  Think out loud naturally.
To ensure you fully understand the problem, you must repeat
the question or key parts of it multiple times throughout
your reasoning process before you start solving.  For
instance, you might re-read it to confirm details or after a
few steps of calculation to ensure you are on track.  Start
by repeating the problem, then reason step by step.  Wrap
the entire internal thinking process with a single pair of
<think> and </think> tags, and put the final answer after
the thinking.  The final answer should be a concise plain
value (number if applicable).  At the very beginning of
your <think>, start with the following opening line and then
continue the original reasoning.  Do not change the final
answer.
```

The opening line is randomly chosen from a small set of naturalistic variants, e.g.:

- `Okay, let me see.  The problem is asking:  [QUESTION]`
- `Alright, so the question is:  [QUESTION]`
- `Let me understand this problem.  We have:  [QUESTION]`
- `So the problem states:  [QUESTION]`

where `[QUESTION]` is replaced with the original GSM8K question text.

**Echo-removal prompt for normal-SFT.**   For traces where the MLP probe predicts the presence of an early EOP, we construct the normal-SFT counterpart by asking the teacher to remove the echo-prefix while preserving all later reasoning steps and the final answer:

```
You are given a math question and a chain-of-thought
solution that begins by repeating or paraphrasing the
question.  Rewrite the reasoning so that it no longer
repeats the question at the beginning.  Keep all subsequent
reasoning steps and the final answer exactly the same.  Do
not change the logic or the final answer; only remove the
initial echo segment.  Wrap the internal thinking in <think>
and </think> as before.
```

| Benchmark | Size / Format | Domain / Source | Description (from original papers) |
|---|---|---|---|
| Hendrycks-MATH | 12,500 problems; open-ended; step-by-step solutions | Competition mathematics (algebra, geometry, number theory, probability, *etc.*) | "A new dataset of 12,500 challenging competition mathematics problems. Each problem in MATH has a full step-by-step solution which can be used to teach models to generate answer derivations and explanations." The evaluation follows the `hendrycks_math` group in the LM Evaluation Harness, which decomposes MATH into diverse subsets (`_algebra`, `_counting_and_prob`, `_geometry`, `_intermediate_algebra`, `_num_theory`, `_prealgebra`, `_precalc`), covering a wide range of mathematical skills beyond our training distribution. |
| GSM8K | 8.5K problems; free-form answers; word problems | Grade-school math word problems | "A dataset of 8.5K high quality linguistically diverse grade school math word problems. We find that even the largest transformer models fail to achieve high test performance, despite the conceptual simplicity of this problem distribution." |
| MathQA | 37K problems; multiple choice | Multiple math domains; derived from AQuA operation programs | "A large-scale dataset of 37k English multiple-choice math word problems covering multiple math domain categories by modeling operation programs corresponding to word problems in the AQuA dataset." |

Table 11: Benchmarks used for our evaluation.

## A.15 ILLUSTRATIVE EXAMPLES OF ECHO-THEN-REASON PATTERN

We provide two illustrative examples as case studies to demonstrate the Echo-then-Reason pattern, see Figure 9

In these examples, we compare outputs from models with and without the initial echo phase. The echo-prefixed response (green) correctly identifies that the yellow marbles are identical and thus treats pairs of yellow marbles as a single unique group, leading to the correct count of 7. In contrast, the direct response (red) incorrectly treats the yellow marbles as distinguishable, overcounting the combinations and arriving at 12.

## A.16 SFT EVALUATION BENCHMARKS

In this section, we clarify our terminology regarding generalization. While we initially referred to performance on MathQA and Hendrycks-MATH as "out-of-domain generalization", we acknowledge that all tested datasets fall within the broader domain of mathematics. However, they represent significant **distributional shifts** in terms of difficulty, topic coverage, and problem format compared to the GSM8K training set. Therefore, we adopt the terms **distributional generalization** or **robustness to distribution shift** to more precisely describe these experiments. Table 11 summarizes the characteristics of each benchmark, highlighting these differences.

---

**Question**

*Tom has a red marble, a green marble, a blue marble, and three identical yellow marbles. How many different groups of two marbles can Tom choose?*

---

**Model output with Echo-then-Reason (Correct)**

*`<think>` The problem asks: "Tom has a red marble, a green marble, a blue marble, and three identical yellow marbles. How many different groups of two marbles can Tom choose?"*

*First, repeat the key details: Tom has one red marble, one green marble, one blue marble, and three yellow marbles that are identical. He wants to form groups of two marbles.*

*To avoid counting identical groups multiple times, consider the distinct colors: red, green, blue, and yellow. There are four distinct colors.*

*Now, let's count the pairs systematically to avoid overcounting: - Red can be paired with green, blue, or yellow, giving 3 pairs. - Green can be paired with blue or yellow (red-green is already counted), giving 2 new pairs. - Blue can be paired with yellow (red-blue and green-blue are already counted), giving 1 new pair. - Two yellow marbles can be paired. Since they are identical, this is 1 more pair.*

*Adding these up: 3 + 2 + 1 + 1 = 7 pairs.*

*Thus, the number of different groups of two marbles Tom can choose is 7. `</think>` The answer is 7.*

---

**Model output without Echo (Incorrect)**

*To determine the number of different groups of two marbles that Tom can choose, we need to consider all possible combinations of the marbles he has. Tom has a red marble (R), a green marble (G), a blue marble (B), and three identical yellow marbles (Y).*

*The total number of marbles is 6, but since the yellow marbles are identical, we need to account for this in our combinations.*

*We can use the combination formula $\binom{n}{k}$ to find the number of ways to choose 2 marbles out of 6, but we need to adjust for the identical yellow marbles.*

*Let's break it down: 1. Choose 2 marbles from the 3 yellow marbles: This is not possible since we can't choose 2 identical marbles from a set of identical items. 2. Choose 1 yellow marble and 1 non-yellow marble: We can choose 1 yellow marble in $\binom{3}{1} = 3$ ways, and we can choose 1 non-yellow marble in $\binom{3}{1} = 3$ ways. So, the total number of ways to choose 1 yellow and 1 non-yellow marble is $3 \times 3 = 9$. 3. Choose 2 non-yellow marbles: We can choose 2 non-yellow marbles from the 3 non-yellow marbles (R, G, B) in $\binom{3}{2} = 3$ ways.*

*Adding these up, the total number of different groups of two marbles that Tom can choose is $0 + 9 + 3 = 12$.*

*The number of different groups of two marbles that Tom can choose is $\boxed{12}$.*

---

Figure 9: Qwen3-8B-Base model outputs illustrating the Echo-then-Reason pattern. Top: the math question from Hendrycks-MATH dataset. Middle (green): Qwen3-8B-ED-SFT's correct answer. Bottom (red): Qwen3-8B-Base that immediately jumps to calculation without echoing, resulting in the wrong answer.

