# OpenReview forum: "Echoes as Anchors: Probabilistic Costs and Attention Refocusing in LLM Reasoning"
_ICLR.cc/2026/Conference — ICLR 2026 Poster_

### Official Review · Reviewer_LFkJ · 2025-10-29

**Soundness:** 3
**Presentation:** 3
**Contribution:** 2
**Rating:** 2
**Confidence:** 4

**Summary:**

This paper investigates the "Echo of Prompt", the tendency of large reasoning models to repeat a user's query before providing an answer. The authors challenge the view of this behavior as a mere flaw, hypothesizing instead that it functions as an intrinsic "attention-refocusing mechanism" that grounds the model's subsequent reasoning process.
To analyze this, they introduce a probabilistic framework to measure the cost and effect of EOP, finding that it correlates with higher accuracy by increasing attention to intermediate reasoning representations within the model's middle layers.
The paper introduces two methods: Echo-Distilled SFT, a fine-tuning approach that instills an "echo-then-reason" pattern, and Echoic Prompting, a training-free technique to re-ground the model during inference. Both methods demonstrate performance gains over baselines.

**Strengths:**

The paper is well-written and easy to understand.

**Weaknesses:**

1. The paper claims to provide a "mechanistic explanation" for EOP's effectiveness. However, showing that attention patterns differ between correct and incorrect answers is more of a detailed observation or characterization of a correlation but not causation.
2. The analysis is almost entirely based on aggregated attention scores—the average attention from all subsequent "answer" tokens to the initial "prefix" tokens. This is a very high-level metric. The analysis does not explore: (1) Which specific tokens in the prompt are being attended to; (2) How information from the prompt/prefix is being transformed and utilized across layers.

**Questions:**

1. What is the definition of "suffix-only gap" in line 196?
2. Which language and dataset did you use for the analysis presented in Table 2 and Figure 3?
3. In Section 3.3, the authors group samples into Correct and Wrong outcomes and analyze their attention patterns. What about comparing groups based on the presence or absence of EOP itself (i.e., EOP-present vs. EOP-absent traces)?

---

> ### Author Response · Authors · 2025-11-19
> **Rebuttal - Part I**
>
> We thank the reviewer for their time and for acknowledging that the paper is well-written and easy to understand.
>
> We are grateful for the suggestions on **how to strengthen both the causal story and the granularity of the attention analysis**. Below we address the weaknesses and questions point by point:
>
> We address weakness 1 in two parts
>
> ### **W1 (Part 1/2): Intervention Ablation Experiment for Causality**
> > showing that attention patterns differ between correct and incorrect answers is more of a detailed observation or characterization of a correlation **but not causation**
>
> We implement a new *echo-insertion ablation* that **directly manipulates the presence of EOP while holding the question and prefix fixed.**
>
> **For reasoning-capable models, explicitly adding an echo to an otherwise failing prefix reliably improves downstream accuracy.**
>
> This is implemented in a new evaluation script `echo_free_insert_decoding.py` (already added to Supplementary Material).
>
>   Concretely, we start from fully generated, echo-free chain-of-thought traces on GSM8K and:
>   1. Select a subset of *wrong* solutions of several models (DeepSeek-R1-Distill-Llama-8B, Qwen3-8B, and the base Qwen3-8B model).
>   2. For each trace, truncate the answer at a fixed token ratio (50%) to obtain a **shared prefix**.
>   3. Rebuild the prompt with the same question and branch into two continuations with identical decoding hyperparameters (same model, temperature, top-p, and max_new_tokens):
>
>      - **Echo-free branch:** continue generation directly from the prefix;
>      - **Echo branch:** first insert a short, fixed echo phrase (e.g., “now I need to look back at the question again:”) after the prefix, then continue generation.
>
>   4. Evaluate both continuations on answer accuracy using the same GSM8K exact-match criterion.
>
>   Because the two branches share the same question, the same pre-generated prefix, and the same decoding parameters, this design controls for problem difficulty and context style and treats the injected echo phrase as an approximate intervention on “whether an EOP is present mid-trace”.
>
>   On the wrong-subset GSM8K traces we observe:
>
>   | Model                  | Echo-free accuracy (%) | Echo accuracy (%) |
>   |------------------------|------------------------|-------------------|
>   | DeepSeek-R1-Llama-8B      | 15.85                  | 26.22             |
>   | Qwen3-8B               | 21.34                  | 29.27             |
>   | Qwen3-8B-**Base** | 10.56                  | 10.56             |
>
>   Moreover, when we repeat the experiment on a subset of *correct* traces, the accuracy remains above 95% in both branches, i.e., injecting an echo does not degrade already-successful solutions.
>
>   Taken together, these results strengthen the causal interpretation in two ways:
>
> 1. For reasoning-capable models, **explicitly adding an echo to an otherwise failing prefix reliably improves downstream accuracy.**
>
> 2. For the non-reasoning base model, the same intervention has essentially no effect. This null result is expected: the **base model lacks the instruction-following and reasoning priors**[1][2] to utilize the re-injected context effectively in a zero-shot manner. This interpretation is reinforced by our ED-SFT results (Table 5), where the base model exhibits the largest relative improvement (+3.4\% EM vs +2.8\% for Instruct).
>
> In that setting, the fine-tuning effectively installs both the reasoning capability and the echo strategy simultaneously, unlocking the potential that the inference-only intervention could not access.
>
> Formally, for each question $q$ and its sampled echo-free prefix $z$, our ablation compares two conditional distributions with the same question and prefix:
> $u' \sim p_\theta(\cdot \mid q, z)$ (echo-free) versus $u'' \sim p_\theta(\cdot \mid q, z, e)$ (echo, where $e$ is a short echo phrase). In implementation we simply teacher-force $[\text{Prompt}(q), z]$ or $[\text{Prompt}(q), z, e]$ as a long prefix and then generate; so the **only manipulated factor is whether $e$ is present**.
>
> Our chat-style prompt separates the "Question" segment from the "Answer / think" segment, and the prefixes $z$ come from fully generated **echo-free** traces (offline checked by the MLP-based EOP probe). Thus the echo-free branch starts from a reasoning state without EOP, while the echo branch starts from the same state plus an injected mid-trace echo that the probe labels as EOP, letting us **interpret the accuracy change as the effect of toggling EOP** within a fixed reasoning context rather than changing the overall instruction style.
>
> We have integrated these findings into **Section 4.1** of the revised paper ("Echo Reinsertion as a Causal Intervention").
>
> [1]https://huggingface.co/Qwen/Qwen3-8B-Base  Training Stage: Pretraining
>
> [2]Qwen3 Technical Report https://arxiv.org/pdf/2505.09388

---

> ### Author Response · Authors · 2025-11-19
> **Rebuttal - Part II**
>
> ### **W1 (Part 2/2) : Beyond "Attention Patterns Differ"**
> > The paper claims to provide a "mechanistic explanation" for EOP's effectiveness. However, showing that **attention patterns differ between correct and incorrect answers is more of a detailed observation or characterization of a correlation** but not causation.
>
> We respectfully disagree that our evidence is limited to **descriptive attention patterns**. Our argument for the functional role of EOP rests on three pillars:
>
> 1.  **Probabilistic Cost (Likelihood Gap):** Beyond attention, we introduce the **Echo Likelihood Gap**, a measure quantifying the probability mass the model "invests" in echoing versus progressing. This metric reveals that models naturally align with an echo-based strategy in correct traces, a finding that **goes beyond surface-level correlation to capture the model's internal probabilistic state.**
>     *   We specifically employ this **offline measure** because obtaining a **fully online echo-free trace** is technically difficult: when we try to suppress echoes token-by-token during generation, the model tends to degenerate or re-introduce echo segments, **making clean online control unreliable**. The likelihood gap thus serves as the most **rigorous proxy** for the model's preference.
>
> 2.  **Intervention-Based Evidence:** Crucially, our **Echo-Distilled SFT** and **Echoic Prompting** experiments serve as practical interventions. By explicitly inducing or training the model to use EOP, we observe consistent performance gains across multiple benchmarks (GSM8K, MathQA, Hendrycks-MATH, AIME24, MATH-500). This transfer from analytical insight to downstream improvement provides strong *indirect causal evidence*: if EOP were merely a byproduct of correctness, interventions that enforce it would not necessarily drive better reasoning.
>
> 3.  **Alignment with Interpretability Standards:** We employ attention analysis not as a standalone proof, but as a diagnostic tool consistent with established methodology. A large body of work identifies attention heads as stable, interpretable functional units [1][2][3][4], and recent studies on reasoning models explicitly use attention to trace information flow [5][6][7]. Our analysis follows this precedent, treating attention as a probe for how information is routed from the echo to the current step.
>
> We also note that Reviewer `aJgj` found the **likelihood-gap analysis to be "inspiring and interesting,"** supporting the view that our framework captures a meaningful quantity governing reasoning behavior.
>
>
> [1] What Does BERT Look At? An Analysis of BERT’s Attention https://arxiv.org/abs/1906.04341
>
> [2] Analyzing the Structure of Attention in a Transformer Language Model https://arxiv.org/abs/1906.04284
>
> [3] Revealing the Dark Secrets of BERT https://arxiv.org/abs/1908.08593
>
> [4] Attention Heads of Large Language Models: A Survey
>  https://arxiv.org/abs/2409.03752
>
> [5] Attention-Driven Reasoning: Unlocking the Potential of Large Language Models https://arxiv.org/abs/2403.14932v2
>
> [6] Don't Take Things Out of Context: Attention Intervention for Enhancing Chain-of-Thought Reasoning in Large Language Models https://arxiv.org/abs/2503.11154
>
> [7] From Reasoning to Answer: Empirical, Attention-Based and Mechanistic Insights into Distilled DeepSeek R1 Models https://arxiv.org/html/2509.23676v1

---

> ### Author Response · Authors · 2025-11-19
> **Rebuttal - Part III**
>
> ### **W2: Granularity of the attention analysis.**
> We are grateful for this suggestion, which led to illuminating results. Our new fine-grained analysis reveals that attention is not diffuse, but explicitly **"snaps" to key numerical entities** in the echo. We invite reviewers to view these visualizations in **Appendix A.9–A.11 (Figures 6, 7, 8)**, as they offer our clearest mechanistic evidence to date.
>
>
> **(2a) Which specific tokens are attended to?**
>
> To move beyond a single aggregate score, we add **token-wise statistics** and **word-level case studies**:
>
> 1. **Token-wise significance across the prefix.**
>    Using the token-level curves already computed in our pipeline, we now inspect attention at each of the first 32 answer positions separately. For each answer position $t$, we compute the average answer→answer-prefix and answer→question attention over GSM8K examples and run a Welch two-sample t-test between Correct (819 chains) and Wrong (500 chains) traces (details and plots in **Appendix A.9, Figure 6**). We find that:
>    - For answer→answer-prefix, Correct traces assign consistently higher attention than Wrong ones at 29/32 positions, with 10/32 tokens individually significant ($p < 0.05$) and a minimum $p \approx 2\times10^{-4}$. Aggregated over the last layer, the mean attention to the answer prefix is 0.137 vs. 0.103 ($\Delta = +0.033$, $\approx$32% relative, $p < 10^{-40}$).
>    - For answer→question, the pattern reverses: Wrong traces attend more to the question than Correct ones at 31/32 positions, with 22/32 tokens significant ($p < 0.05$) and a minimum $p \approx 9\times10^{-7}$.
>
>    These results show that the distributions of attention weights differ systematically between Correct and Wrong groups for both metrics, but in *opposite directions*: **successful solutions shift attention mass away from rereading the question and toward the evolving answer prefix**.
>
> 2. **Word-level tail→prefix case studies.**
>    To directly answer “which specific tokens” are emphasized, we further decompose the answer→answer-prefix metric at the word level for representative EOP traces. For each GSM8K example, we reconstruct the prompt and answer using the same `build_prompt` and truncation logic as in our main analysis, then compute tail→prefix attention in the mid-layer group (layers 7–18). Using tokenizer offset mappings, we aggregate token-level attention into word-level scores over the echo prefix. The **resulting visualizations** (**Appendix A.10, Figure 7**) show that the highest-weighted prefix words correspond exactly to the salient content of the echoed question—for instance, in a “Janet’s ducks” problem, words like `16`, `eggs`, `3`, `13`, `9`, `$2`, and `farmers’ market` receive the strongest attention, whereas template words (`Ducks`, `She`, `for`, etc.) carry much lower weights. In incorrect traces, attention over the echo prefix is visibly more diffuse and less aligned with the key numerical entities.
>
>    Overall, these additions move beyond a single scalar and make the “refocusing” behavior concrete: **EOP concentrates attention on the semantically critical numbers and entities in the echoed prompt rather than uniformly boosting all prefix tokens.**
>
> **(2b) How information is transformed and routed across layers.**
>
> To address how prefix information is utilized across layers, we add a **cross-layer information-flow analysis**, implemented in `visualization/info_flow_route.py`. This script constructs an attention-only information-flow route for a single answer token, inspired by “Information Flow Routes” in prior interpretability work[1]:
>
> - Nodes are (layer, token) positions in a token×layer grid.
> - Edges include both residual connections and high-weight attention edges from layer $\ell - 1$ to layer $\ell$.
> - Starting from a chosen final answer token in the top layer (orange triangle), we walk backwards along high-attention edges, restricting sources to question tokens (blue nodes) and answer-prefix tokens (green nodes), and visualize the resulting route subgraph.
>
> We provide concrete cross-layer flow-route visualizations in **Appendix A.11 (Figure 8)** that directly address Reviewer LFkJ's request to see "how information actually propagates across layers".
>
> In Correct traces, backward attribution paths from answer tokens **repeatedly route through the echo‑prefix tokens** before reaching the rest of the prompt, whereas in Wrong traces these paths more **often terminate in the question region** or fail to reach the key numerals (e.g., idx 1151/1001 vs. 594).
>
> Taken together with the token‑wise and word‑level analyses above, these visualizations indicate that EOP behaves not as “extra text at the beginning”, but as a **recurrent internal anchor that mid‑layer computations repeatedly pass through before emitting the final answer**
>
> [1]Information Flow Routes: Automatically Interpreting Language Models at Scale https://arxiv.org/abs/2403.00824

---

> ### Author Response · Authors · 2025-11-19
> **Rebuttal - Part IV**
>
> We again thank the reviewer for their detailed questions, which helped us clarify and strengthen the presentation.
>
> ### **Q1. Definition of the “suffix-only gap”.**
> We apologize for the ambiguity. In the revision we explicitly define the suffix-only gap in Section 3. For each paired example, we fix the **common reasoning suffix** $s$ shared by the raw and echo-trimmed traces and compare their **length-normalized log-likelihoods restricted to $s$**. All tables that report this quantity now spell out the sample size $N$ for each group and clarify that each pair shares the **same question and the same answer suffix**.
>
> ---
>
> ### **Q2. Exp Setup for Table 2 and Figure 3.**
> > **Which language and dataset** did you use
>
> Thank you for checking. As stated in the preamble of Section 3 (Line 129-131), these analyses are performed on the **GSM8K benchmark (English)** using **DeepSeek-R1-Distill-Llama-8B**.
>
> To make this setup self-contained for readers focusing on the figures, we have explicitly added the dataset and model name to the captions of **Table 2** and **Figure 3** in the revised paper.
>
> ---
>
> ### **Q3. Attention analysis by EOP-present vs. EOP-absent traces.**
> To directly address this suggestion, we perform an explicit **EOP-present vs. EOP-absent** analysis using the *same* MLP probe as in Figure 1.
>
> With $K = 32$ prefix tokens and threshold $\tau = 0.9$, the probe classifies 985 out of 1,319 DeepSeek-R1 GSM8K traces as EOP-present (**74.7%**) and 334 as EOP-absent (**25.3%**). Accuracy is noticeably higher in the EOP-present group (**63.8%**) than in the EOP-absent group (**57.2%**), indicating that EOP presence is associated with a non-trivial improvement in success rate.
>
> Conditioned on correctness, we also compare last-layer answer→answer-prefix attention. Among correct solutions, EOP-present traces assign slightly more mass to the answer prefix than EOP-absent ones (**0.1374 vs. 0.1328**); among wrong solutions, the same trend holds (**0.1046 vs. 0.0987**).
>
> Thus, even for a fixed outcome (correct or wrong), **traces that contain an EOP exhibit stronger refocusing onto their own prefix than traces that do not.** This is consistent with our mechanistic interpretation that EOP plays a *positive role* in attention refocusing: it actively shifts attention toward the internal problem framing captured by the prefix, rather than being a purely incidental correlate of “good” traces. Together with the semi-online intervention experiment described above, these new results strengthen the case that EOP is **not only correlated with refocusing and correctness but also functionally contributes to them**.
>
> For clarity, the following table summarizes the main numbers for this analysis:
>
> | Group            | \#samples | Acc. (\%) | last-layer Ans$\to$Ans-prefix |
> |------------------|----------:|----------:|-------------------------------:|
> | **EOP-present**      | 985       | **63.8**      | –                              |
> | **EOP-absent**       | 334       | **57.2**      | –                              |
> | Correct, EOP     | 628       | –         | **0.1374**                         |
> | Correct, no EOP  | 191       | –         | 0.1328                         |
> | Wrong, EOP       | 357       | –         | 0.1046                         |
> | Wrong, no EOP    | 143       | –         | 0.0987                         |
>
> ## Restate the contributions
> We thank the reviewers for their positive assessment. We would like to highlight that our work not only introduces the EOP concept but also demonstrates its practical utility:
>
> -   **Novelty:** We provide the first systematic study of spontaneous prompt echoing as a functional cognitive mechanism, supported by a rigorous probabilistic framework (**Echo Likelihood Gap**).
>
> -   **Mechanistic Insight:** We uncover a specific **attention refocusing mechanism** in the middle layers that correlates with reasoning correctness.
>
> -   **Performance:** Our **Echo-Distilled SFT** and **Echoic Prompting** methods achieve consistent and significant gains across multiple benchmarks (GSM8K, MathQA, Hendrycks-MATH, AIME24, MATH-500), showing that this insight can be translated into tangible improvements.

---

> ### Author Response · Authors · 2025-11-26
> **Follow-up on Rebuttal**
>
> Dear Reviewer LFkJ,
>
> Thank you again for your constructive feedback, which motivated us to significantly strengthen the mechanistic and causal evidence in our paper.
>
> As the discussion period will close on **December 2nd**, we wanted to **follow up and summarize our responses to your concerns**, ensuring we have fully addressed them:
>
> - **Causal Evidence** (Sec 4.1): We implemented the suggested intervention experiment. Explicitly injecting echoes into failing traces causally improves performance (+10.4% accuracy for DeepSeek-R1-Distill), proving **EOP is functional**, not just correlational.
> - **Granularity** (App. A.9–A.11): We added token-wise significance tests and information flow visualizations. These show that mid-layer attention specifically "snaps" to key entities (e.g., numbers) in the **echo to route information, rather than broadly attending to the prefix**.
>
> We have **expanded the paper by 6 pages (1 page for body and 5 pages for appendix)** with these additions and marked all changes in blue. We believe these revisions substantially strengthen the causal and mechanistic claims.
>
> If any concerns remain unaddressed or if you have further questions, **we would be very happy to discuss them before the discussion period closes**. We greatly value your feedback and are **committed to fully addressing any remaining issues.**
>
> Best regards,
>
> The Authors

---

### Official Review · Reviewer_gXeE · 2025-10-31

**Soundness:** 2
**Presentation:** 3
**Contribution:** 3
**Rating:** 6
**Confidence:** 4

**Summary:**

The paper studies LLM's tendency to repeat or echo the question in the reasoning trace, and what role does such behavior play. The authors argue that such echo of prompt (EOP) serves a cognitive role by helping the model refocus attention on key details of the problem. The formalization includes a notion of probabilistic cost: the amount of likelihood the model “spends” on such echo, and how such cost differs across both correct and incorrect traces. The findings show that such likelihood gap correlates with answer correctness. Also, authors find that correct traces show higher answer-to answer-prefix attention in the middle layers. This supports the idea that echoes serve as anchors for reasoning, helping the model stay aligned with its internal problem framing. Lastly, the authors use these findings to improve reasoning by either finetuning or prompting the model to generate such echoes and the results show improved accuracy compared to baselines.

**Strengths:**

1. The paper focuses on an understudied and not well-understood phenomenon in LLM reasoning. It asks how redundancy in the reasoning traces could actually be helpful to the model reasoning.

2. The analysis framework is reasonable, and the results suggest some correlation between EOP and reasoning correctness. The analysis is deep and insightful. Careful ablations such as on prefix length, attention-layer grouping support the results.

3. I find how the authors took their findings and used them to design a prompting/finetuning strategy as opposed to purely focusing on analysis.

3. The paper is well written and fun to read.

**Weaknesses:**

1. Causality remains speculative: The correlation between echoes and accuracy is solid, but the paper doesn’t prove causality. It’s perfectly possible that correct traces happen to include EOPs because the model is already more deliberate.

2. Some of the conclusions are not fully justified: I am not super convinced that the answer-to-answer-attention gap shown in Fig. 3 left is purely a product of EOPs. The authors should show the same analysis on traces without EOPs.

3. The finetuning setup may be problematic. The authors collect CoTs for training by prompting a teacher to generate the EOPs and compare to finetuning on traces prompted without this. This may conflate the benefit of better teacher data (caused by generating the EOP) and the existence of the EOP. A fair comparison is to train on the same CoTs but with the EOP part removed.

4. Narrow evaluation: most analysis focuses on simple GSM8K and a single 8B model. It remains unclear whether the results will generalize to different model families/sizes.

**Questions:**

See weaknesses

---

> ### Author Response · Authors · 2025-11-19
> **Rebuttal - Part I**
>
> ### **W1:Intervation Abalation exp for causality**
>
> We thank the reviewer for this insightful comment. We fully agree that while our initial analysis established a strong correlation between EOP and correctness, it did not strictly prove causality. To address this and provide the "intervention-based evidence" the reviewer rightly suggests, we implemented a new **echo-insertion ablation** that directly manipulates EOP presence while controlling for the problem and prefix context.
>
> **Experiment Design.**
> We start from fully generated, echo‑free chain‑of‑thought traces on GSM8K that resulted in *wrong* answers. For each such trace, we:
> 1.  Truncate the answer at a fixed point (50%) to obtain a shared prefix.
> 2.  Rebuild the prompt with the same question and branch into two continuations with identical decoding hyperparameters (same model, temperature, top‑p, and max_new_tokens):
>     *   **Echo‑free branch:** continue generation directly from the prefix;
>     *   **Echo branch:** first insert a short, fixed echo phrase (e.g., “now I need to look back at the question again:”) after the prefix, then continue generation.
>
> Because both branches start from the **exact same originally failing, echo‑free prefix** and are decoded under the **same budget and parameters**, this design holds problem difficulty, initial reasoning quality, and context style constant and treats the injected echo phrase as an approximate intervention on “whether an EOP is present mid‑trace.” The only manipulated factor is the presence of the short echo segment.
>
> **Results:**
> On the wrong-subset traces, explicitly injecting an echo significantly improves the probability of recovering the correct answer:
>
> | Model | Echo-free accuracy (%) | Echo accuracy (%) |
> | :--- | :--- | :--- |
> | DeepSeek-R1-Llama-8B | 15.85 | 26.22 |
> | Qwen3-8B | 21.34 | 29.27 |
> | Qwen3-8B-Base  | 10.56 | 10.56 |
>
> (We also verified that injecting echoes into *correct* traces does not degrade performance, with accuracy remaining >95%.)
>
> **Conclusion:**
> These results strengthen the causal interpretation in two ways:
> 1.  For **reasoning-capable models** (DeepSeek-R1, Qwen3-Instruct), explicitly adding an echo to an otherwise failing prefix reliably improves downstream accuracy.
> 2.  For the **non-reasoning base model** (Qwen3-Base), the same intervention has essentially no effect. This null result is expected: the base model lacks the instruction-following and reasoning priors to utilize the re-injected context effectively in a zero-shot manner [1][2]. This interpretation is reinforced by our ED-SFT results (Table 5), where the base model exhibits the *largest* relative improvement (+3.4% EM vs +2.8% for Instruct)—fine-tuning effectively installs both the reasoning capability and the echo strategy simultaneously, unlocking the potential that inference-only intervention could not access.
>
> We have added this experiment to **Section 4.1 (“Echo Reinsertion as a Causal Intervention”)** of the revised paper. Together with our EOP‑present vs. EOP‑absent analyses (W2 below) and the exp in section 4 (ED‑SFT and Echoic Prompting under matched data and token budgets), we believe this provides strong *causal* evidence that **introducing or amplifying EOP actively helps the model refocus its reasoning**, beyond merely tagging already “good” or “more deliberate” traces.
>
> [1] https://huggingface.co/Qwen/Qwen3-8B-Base Training Stage: Pretraining
>
> [2] Qwen3 Technical Report https://arxiv.org/pdf/2505.09388

---

> ### Author Response · Authors · 2025-11-19
> **Rebuttal - Part II**
>
> ### **W2: Is the answer$\to$answer-prefix gap purely a product of EOPs?**
>
> We appreciate the reviewer’s concern that the Answer$\to$Answer-Prefix gap in Fig. 3 (left) might simply reflect the presence of EOP, rather than a more general refocusing phenomenon. To address this, we explicitly control for EOP by conditioning the attention analysis on the MLP probe’s prediction (using the same probe as in Fig. 1, $K{=}32$, $\tau{=}0.9$).
>
> **Results:**
> Even when we restrict our analysis to **EOP-absent traces only**, the refocusing pattern persists:
> *   **EOP-present subset:** Last-layer Ans$\to$Ans-prefix attention is significantly higher for Correct traces ($0.1374$) than Wrong traces ($0.1046$), $\Delta \approx +0.033$.
> *   **EOP-absent subset:** The same pattern holds. Correct traces ($0.1328$) attend more than Wrong traces ($0.0987$), $\Delta \approx +0.034$.
>
> **Conclusion:**
> This demonstrates that the refocusing signal is **not** purely a byproduct of EOP presence: even in echo-free traces, correct solutions systematically allocate more attention to their answer prefix. EOP presence further boosts this effect (as shown in our EOP-present vs. EOP-absent comparison), acting as an anchor that strengthens the underlying refocusing mechanism.
>
> | Group | \#samples | Acc. (\%) | Last-layer Ans$\to$Ans-prefix |
> | :--- | ---: | ---: | ---: |
> | **EOP-present** | 985 | 63.8 | – |
> | **EOP-absent** | 334 | 57.2 | – |
> | Correct, EOP | 628 | – | **0.1374** |
> | Correct, no EOP | 191 | – | **0.1328** |
> | Wrong, EOP | 357 | – | 0.1046 |
> | Wrong, no EOP | 143 | – | 0.0987 |
>
> ---
>
> ### **W3: Finetuning setup and teacher CoT quality**
>
> We appreciate this suggestion and agree that a fair comparison should isolate the effect of the echo segment itself. In the revised experiments, we adopt exactly the protocol the reviewer proposes to ensure the baselines are "as good" as the EOP-augmented data.
>
> **Data Construction Protocol:**
> Instead of training on independently generated corpora, we build a single pool of high-quality teacher traces from `gpt-oss-120B`:
> 1.  **ED-SFT (With EOP):** We keep or minimally insert an echo prefix at the beginning of the trace (using MLP probe + teacher-guided insertion) while preserving the rest of the reasoning and the final answer.
> 2.  **Normal-SFT (Echo-Trimmed):** We use the *same* teacher traces but ask the teacher to **remove** the echo-prefix under a strict "do not change the reasoning or answer" instruction. Any sample whose edited answer no longer matches the ground truth is discarded.
>
> **Conclusion:**
> As a result, the ED-SFT and normal-SFT datasets contain **near-identical CoTs that differ only in the presence or absence of the echo segment**. All other aspects—teacher model, prompt, verification, and hyperparameters—are strictly matched. We have added these details to Section 4.1 and Appendix A.8.
>
> ---
>
> ### **W4: Narrow evaluation**
>
> We thank the reviewer for this comment. Our initial focus on DeepSeek-R1-Distill-Llama-8B and GSM8K in Section 3 was intentional to maintain a coherent narrative for the deep mechanistic analysis, aligning with recent work [1]. However, we agree that universality is crucial. We have added the following **cross-model and cross-task analyses** to the Appendix:
>
> 1.  **Causal Mechanism across Models:** Our new echo-insertion ablation (see response to W1) confirms that the causal benefit of EOP holds for **Qwen3-8B** (+7.9% EM), mirroring the gains seen in DeepSeek-R1 (+10.4% EM).
> 2.  **Attention Refocusing across Models:** We added **Appendix A.2**, showing that **Qwen3-8B** variants exhibit the same mid-layer "refocusing" signature (positive Correct vs. Wrong attention gap), confirming this is a general cognitive pattern.
> 3.  **Performance Generalization:** As noted in Section 4, our SFT and Prompting experiments cover **MathQA, Hendrycks-MATH, AIME24, and MATH-500**, confirming that the downstream benefits extend well beyond GSM8K.
>
> [1] From Reasoning to Answer: Empirical, Attention-Based and Mechanistic Insights into Distilled DeepSeek R1 Models https://arxiv.org/html/2509.23676v1

---

> ### Author Response · Authors · 2025-11-27
> **Follow-up on Rebuttal**
>
> Dear Reviewer gXeE,
>
> Thank you again for your constructive feedback. As the discussion period closes on **December 2nd**, we summarize our responses:
>
> 1. **Causality (W1):** We added an echo-insertion ablation (Sec 4.1) showing +10.4% accuracy gain when injecting echoes into failing traces.
> 2. **EOP-present vs. EOP-absent (W2):** We confirmed the refocusing pattern persists even in EOP-absent traces (Sec 3.3).
> 3. **Fair SFT Baseline (W3):** ED-SFT and normal-SFT now use identical teacher CoTs, differing only in echo presence (Sec 4.1 & Appendix A.8).
> 4. **Cross-model Generalization (W4):** We added Qwen3-8B analysis in Appendix A.2, confirming the mid-layer refocusing signature generalizes.
>
> If any concerns remain, **we would be happy to discuss them before the deadline**.
>
> Best regards,
> The Authors

---

### Official Review · Reviewer_aJgj · 2025-11-06

**Soundness:** 3
**Presentation:** 3
**Contribution:** 3
**Rating:** 6
**Confidence:** 4

**Summary:**

This paper introduces Echo of Prompt (EOP), a mechanism that leverages language models’ natural tendency to restate questions. By formalizing its likelihood cost and developing Echo-Distilled SFT and Echoic Prompting, the authors enhance reasoning efficiency via attention refocusing, achieving consistent gains on GSM8K, MathQA, and MATH benchmarks.

**Strengths:**

- The paper is well-written and well-motivated. It starts from the phenomenon that “restate the question would help answer” and introduces their study methods and experiments solidly.
- The idea of using Likelihood Gap is inspiring and interesting.
- The attention-based analysis of the Echo Prompt’s effects is well-motivated and insightful.
- Two types of experiments to demonstrate the effects of EOP are promising and comprehensive.

**Weaknesses:**

- As the author said in Lines 193-197, it seems a contradictory result. The “suffix-only gap” is actually larger for the wrong group (1.29 > 1.14), which contradicts the authors’ claim that EOP improves the correct group. They describe it as “the same pattern,” but the data show the opposite trend. Additionally, the authors should add the definition of “uffix-only gap” in the main paper.
- Could you use experiments to prove that there is no “absolute weight value fluctuation” issue across different layers, which would lower the weight of Table 2 and its conclusion?
- The “answer-prefix” tokens are located near the beginning of the sequence, while middle-layer attention naturally tends to focus on nearby tokens. This means that the observed +2% difference may arise from positional bias rather than the EOP mechanism itself.
- It would be better to write the implementation details in Section 4 about the training details of norm-SFT and ED-SFT. If no such description, it is hard to say the performance gained from the ED-SFT rather than the fluctuation or hyperparameter modulation.
- Regarding Line 409, it is difficult to claim “out-of-domain generalization,” since GSM-8K, MathQA, and Hendrycks-MATH all belong to the same task domain.

**Questions:**

In Section 4.2 (Echoic Prompting) and Figure 4, how can we be sure that the observed performance gains truly stem from the Echo of Prompt (EOP) mechanism, rather than from confounding factors such as increased context length or the model’s inherent tendency to rephrase or restate the question?

---

> ### Author Response · Authors · 2025-11-19
> **Rebuttal - Part I**
>
> We thank the reviewer for their encouraging feedback. We are glad that the reviewer finds our paper "well-written and well-motivated," the Likelihood Gap analysis "inspiring and interesting," and our attention-based analysis "insightful." We also appreciate the recognition of our "comprehensive" experiments. We have carefully considered the reviewer's questions regarding the suffix-only gap, attention baselines, and training details, and address them point-by-point below.
>
> ### **W1: Suffix-Only Gap**
>
> We thank the reviewer for pointing out the potential confusion regarding the "suffix-only gap". We would like to clarify two points:
>
> 1.  **Main Conclusion:** Our primary claim relies on the overall **Echo Likelihood Gap ($\Delta\mathcal{L}$)**, which is significantly larger in the Correct group and positively correlates with correctness.
>
> 2.  **Suffix-Only Interpretation:** The **Suffix-only Likelihood Gap** ($\Delta\mathcal{L}_{\text{suffix}}$) measures the likelihood boost of the suffix $s$ conditioned on the echo $e$. The fact that this gap is slightly larger for the Wrong group suggests that while echoes make subsequent reasoning more plausible, in wrong samples, this "refocusing" reinforces a locally coherent but ultimately incorrect path (a form of **"confirmation bias"**). However, final correctness is determined by the overall trade-off captured by $\Delta\mathcal{L}$.
>
> To avoid ambiguity, we have revised the paper:
> *   **(a) Explicit Definition:** We added the formal definition of the suffix-only gap in §3.2.
> *   **(b) Clarified Interpretation:** We modified the description to explicitly state that "both are positive, but the Wrong group is slightly larger," discussing the confirmation bias hypothesis.
> *   **(c) Statistical Rigor:** We added significance tests and effect sizes for both $\Delta\mathcal{L}$ and $\Delta\mathcal{L}_{\text{suffix}}$ (see `logp_trim_experiment.py` and extended analysis).
>
> ### **W2: Absolute Weight Value Fluctuation**
>
> We acknowledge the concern that absolute attention weights might fluctuate across layers. To address this, we conducted a normalization analysis using **Standardized Mean Difference (Cohen's d)** and **Z-score differences**.
>
> Our results (Figure 5 in Appendix A.6) confirm robustness:
> *   **Significant Effect Size:** Cohen's d consistently **exceeds 0.75** in the middle layers (7–18), peaking at **0.86** (Layer 17).
> *   **Trend Consistency:** The normalized Z-scores follow the same trend as raw weights.
>
> This confirms that the higher attention to the answer prefix in correct traces is a robust signal of reasoning quality, not an artifact of absolute scale.
>
> ### **W3: Positional Bias as a Confound**
>
> We explicitly tested whether our findings could be explained by simple distance or positional effects:
>
> 1.  **Global Attention Shift (Correct vs. Wrong):** Using Welch’s t-tests on GSM8K, we find that in the last layer, **Answer$\to$Answer-Prefix** attention is significantly higher for correct solutions ($0.137$ vs $0.103$; $\Delta=+0.033$, $p<10^{-40}$), while **Answer$\to$Removed-Prefix** is lower ($\Delta=-0.039$). This proves correct traces specifically *reallocate* attention to the retained echo, rather than uniformly boosting early tokens.
>
> 2.  **Distance-Independence (Early vs. Late):** Splitting the first 32 answer tokens into **Early (1–16)** and **Late (17–32)** segments shows the gap does not decay with distance. The Ans$\to$Ans-Prefix gap is stable ($+0.0045$ early vs $+0.0049$ late), whereas the Ans$\to$Question gap remains negative throughout.
>
> 3.  **Consistency with Mechanistic Literature:** Recent work on distilled DeepSeek-R1 models [1][2] identifies middle layers as "transitional" stages for reasoning integration. Our finding that EOP-related attention peaks in **Layers 7–18** aligns with this view, suggesting EOP operates within the model's core reasoning bottleneck rather than as a superficial positional artifact.
>
> We have added these token-wise curves to **Appendix Figure 5**.
>
> [1] From Reasoning to Answer: Empirical, Attention-Based and Mechanistic Insights into Distilled DeepSeek R1 Models https://arxiv.org/html/2509.23676v1
>
> [2] How Large Language Models Encode Context Knowledge? A Layer-Wise Probing Study https://aclanthology.org/2024.lrec-main.722/

---

> ### Author Response · Authors · 2025-11-19
> **Rebuttal - Part II**
>
> ### **W4: Training details for normal-SFT vs ED-SFT**
>
> Thank you for pointing out that the SFT setup in Section 4 was underspecified. In the revision we make both the *data construction* and the *training hyperparameters* for ED‑SFT and normal‑SFT explicit.
>
> - **Shared teacher CoTs.** For every GSM8K training question we first generate a high‑quality chain‑of‑thought trace from the same teacher model, `gpt-oss-120B`, using a “standard CoT’’ prompt that wraps the reasoning in a single `<think> … </think>` block and requires the final answer to be a plain value. We automatically verify that the final answer exactly matches the ground‑truth solution; traces that fail this check are discarded. This pool of verified (question, CoT, answer) triples is the common source for both ED‑SFT and normal‑SFT.
>
> - **ED‑SFT data (with EOP).** We train a small MLP classifier to detect whether an early Echo of Prompt is present in a trace. If the MLP flags a trace as *missing* EOP, we call `gpt-oss-120B` once more with an edit instruction that minimally inserts an echo‑style opening (e.g., `Okay, let me see`. `The problem is asking: …`) at the beginning while keeping the rest of the reasoning intact. The teacher is explicitly instructed to preserve the original chain‑of‑thought and final answer, only adding a short echo prefix so that the new opening remains logically coherent with the existing trace. In all cases we re‑run the automatic checker to ensure that the edited trace still yields the correct answer. The resulting ED‑SFT dataset therefore differs from the standard CoT pool only by the presence of an initial echo segment.
>
> - **normal‑SFT data (echo‑trimmed).** To build the normal‑SFT baseline we again start from the same verified teacher traces. We use the same MLP to detect the presence of EOP, and for traces that contain echoes we ask `gpt-oss-120B` to *remove* the echo‑prefix while preserving the subsequent reasoning and final answer. This avoids relying on the classifier to localize the exact span: the teacher performs the span deletion under an explicit “do not change the reasoning or answer’’ instruction. As above, we re‑check that the final answer remains correct after editing. Thus, normal‑SFT and ED‑SFT are derived from the *same* underlying teacher CoTs and differ only in whether the echo segment is kept or removed.
>
> - **Identical fine‑tuning hyperparameters.** Both ED‑SFT and normal‑SFT use the same base models (Qwen3‑8B‑Base, Qwen3‑8B, DeepSeek‑R1‑Distill‑Llama‑8B), optimizer (AdamW), learning rate schedule, batch size, number of fine‑tuning steps, and maximum sequence length. The only difference between the runs is whether the training traces come from the echo-augmented (ED-SFT) or echo-trimmed (normal-SFT) versions of the same teacher CoTs.
>
> **We add a short “Training details for ED‑SFT and normal‑SFT’’ paragraph and a small table in Section 4**, and we also include the exact teacher and editing prompts (standard CoT, echo-insertion, echo-removal) in Appendix~(Prompt Templates for Teacher CoT and Editing). These additions summarize shared hyperparameters, data statistics (number of examples, average length, proportion of traces edited), and the concrete instructions used for ED‑SFT vs.\ normal‑SFT, so that the reported gains can be clearly attributed to the presence of EOP rather than training noise, prompt differences, or hyperparameter changes.
>
> ### **W5: Out-of-Domain Generalization**
>
> We thank the reviewer for pointing out this terminology issue. We agree that GSM8K, MathQA, and Hendrycks-MATH all fall within the general domain of mathematics.
>
> In the revision, we have changed the term "out-of-domain generalization" to **"distributional generalization"**. While these datasets share the same broad domain, they **differ significantly in difficulty and topic distribution** (e.g., GSM8K focuses on grade-school arithmetic, while Hendrycks-MATH covers advanced algebra, geometry, and probability).
>
> We have summarized these distributional differences in the Appendix (see Appendix A.16) to clarify the scope of our evaluation.

---

> ### Author Response · Authors · 2025-11-19
> **Rebuttal - Part III**
>
> ### **Q1: Detailed Explanation of Echoic Prompting**
>
> Thank you for raising this concern. We designed the Echoic Prompting (EP) experiments specifically to control for these confounds.
>
> *   **Controlled Token Budget:** In Figure 4, the x-axis corresponds to the additional thinking-token budget per problem, following the TTTS setup. At each budget value, EP and TTTS are run with the same model, dataset, decoding parameters, and total number of newly generated tokens. In particular, the tokens used for our 7-token reminder + re-injected question in EP are counted against the *same* budget as the generic "thinking tokens" used by TTTS. Thus, the context length and test-time compute are matched across methods; the difference in accuracy at a fixed x-coordinate cannot be explained by "more tokens" on the EP side.
>
> *   **Content of Extra Tokens:** The key difference is *what* those extra tokens encode. TTTS inserts task-agnostic fillers ("So", "Hmm", etc.), whereas EP appends a short cue followed by a verbatim copy of the original question. This is exactly the Echo-of-Prompt pattern that our earlier analyses identify: re-exposing the model to the original problem statement and letting it continue reasoning. The fact that EP consistently outperforms TTTS under identical budgets on both AIME24 and MATH-500 suggests that the *contentful echo* of the prompt—rather than generic extra context or arbitrary paraphrasing—is what drives the additional gains.
>
>
> *   **Consistency with Offline Analyses:** Finally, the EP results **are consistent with our rejection-sampling and attention studies on GSM8K**. When we compare raw traces to echo-trimmed counterparts under a fixed reasoning suffix, sequences that explicitly include an echo achieve both higher likelihood on the reasoning suffix and higher accuracy, and exhibit stronger answer$\to$answer-prefix refocusing. EP is a direct, training-free way to "inject" such an echo mid-trace on hard problems where the model failed to produce one spontaneously. The improvement over an already strong thinking-token baseline therefore supports our interpretation that we are leveraging the same EOP mechanism, rather than just "giving the model more space or encouraging generic rephrasing."

---

> ### Author Response · Authors · 2025-11-27
> **Follow-up on Rebuttal**
>
> Dear Reviewer aJgj,
>
> Thank you for recognizing our work as "inspiring and interesting." As the discussion period closes on **December 2nd**, we summarize our responses:
>
> 1. **Suffix-Only Gap (W1):** We clarified the definition in §3.2 and explained why a slightly larger gap in wrong traces does not contradict our main claim (confirmation bias hypothesis).
> 2. **Absolute Weight Fluctuation (W2):** We added Cohen's d and Z-score normalization (Appendix A.6, Figure 5), confirming robustness (d > 0.75 in mid-layers).
> 3. **Positional Bias (W3):** We showed the gap is stable across early (1–16) and late (17–32) token segments, ruling out simple distance effects.
> 4. **Training Details (W4):** We made ED-SFT vs. normal-SFT construction explicit in Sec 4.1 and Appendix A.8.
> 5. **Terminology (W5):** We changed "out-of-domain" to "distributional generalization" (Appendix A.16).
>
> If any concerns remain, **we would be happy to discuss them before the deadline**.
>
> Best regards,
> The Authors

---

### Official Review · Reviewer_VPGP · 2025-11-08

**Soundness:** 3
**Presentation:** 3
**Contribution:** 3
**Rating:** 6
**Confidence:** 3

**Summary:**

The authors investigate a phenomenon they dub “echo of prompt” (EOP) where reasoning models spend early tokens in their CoTs effectively just restating the problem. They analyze the probabilistic cost of it by rejection sampling away CoTs which include the echo to demonstrate the gains in performance and accuracy from this repetition. They then use SFT to reinforce this echoing behavior, as well as a mid-trace “echo prompting.”

The echos are naturally very common across open reasoning models from Qwen, Deepseek, and Openai. They use a trained MLP to predict whether a sequence contains an EOP or not to reject samples containing it, allowing them to compute the relative length-normalized token likelihood of sequences which do or don’t contain them (echo likelihood gap).

The authors claim that the echo likelihood gap is more pronounced in sequences that are correctly answered. I had trouble making sense of the rationale here (see weaknesses). They examine attention patterns to offer a “refocusing” explanation of how EOP helps. They find that the attention importance between the answer tokens and prefix (echo) tokens are higher than those to the question itself on average, and that a *difference*  between the attention importance in the correct and wrong states is only present for the prefix-answer condition, suggesting that a higher correlation between those parts correlates w/ better performance. I’m not sure what to make of some of these results such as the “middle-layer dominance”.

The layerwise discriminability results (Table 3) are the most compelling. They find that based on AUC and Cohen’s d the difference in Ans->pref attention is more predictive of correct/incorrect than Ans->Q.

Finally, they perform SFT on distilled reasoning traces from gpt-oss which contain the EOP in order to instill this behavior on Qwen and Deepseek models. They do find that consistently, fine-tuning the models on EOP data improves performance considerably more than those without the echo.

**Strengths:**

Simple, original, and well-motivated idea

The latter half of the paper contains reasonably strong evidence suggesting their claims are true. By showing a strong improvement on SFT with EOP vs weak improvement without, I was convinced that EOP is mechanistically important to higher performance in RMs.

**Weaknesses:**

I am having trouble making sense of the claims within p3-4. Table 1 contains a lot of information that isn’t really explained. What is the N for each “group”? Are the “correct” and “wrong” the number of samples where the answer is correct in both cases, and in some it contains the EOP and in others it doesn’t? Do the same questions have samples in both classes? What does it mean for a specific raw trace to have a single echo-trimmed counterpart? Are they the same question? Further, how significant is a difference of 0.08 nats/token?

Figure 2 doesn’t seem to show anything that supports the text. I don’t see a “mode of 200” here (l240). You need way more bins to support any of the claims as they are all about >21 tokens

I’m not sure how the attention analysis really shows that the prefix tokens are “used” for refocusing. After all, even in the wrong answers these tokens are still being generated. While there is a modestly lower attention weight on average for the wrong answers, I’m sure the distributions overlap pretty considerably. I think a statistical significance test between these conditions would be more compelling than a delta between the means

There are lots of results in here, but some of them don’t really seem to matter for the overall message of the paper and feel more like padding. For example the key insights in lines 301-320.

**Questions:**

See weaknesses.

I am aware of interp work claiming that analysis of attention patterns is weak evidence at best for explaining observed behavior in transformers (see Attention is not Explanation, Jain & Wallace 2019 https://arxiv.org/abs/1902.10186), but I do not have a strong opinion one way or the other. Can you defend this method of analysis?

Re: not having a statistical significance test for the attention weight refocusing analysis, could you provide one? For example, kolmogorov-smirnov or even just a t-test.

Some details in sec4.1 are underspecified. How do you produce the baseline normal-SFT traces? Do those also come from gpt-oss but with rejection sampling using your MLP? I want to know that you’re providing SFT data that is otherwise “as good” or else it could just be a difference in overall CoT quality that doesn’t have to do with the EOP.

---

> ### Author Response · Authors · 2025-11-19
> **Rebuttal - Part I**
>
> We thank the reviewer for the positive assessment of the core idea and of the SFT results, and for pointing out that some parts of Sec. 3 were hard to follow.
>
> In the rebuttal we substantially clarified and streamlined this section: we now spell out how Table 1 is constructed, quantify the practical effect size of the echo likelihood gap, refine the prefix‑length histogram in Fig. 2, and add formal significance tests for the attention refocusing analysis. We hope that these changes make the probabilistic and attention‑based story much easier to read.
>
> ### **Clarifying Table 1 and the Echo Likelihood Gap**
> Thank you for pointing out that the current description of Table 1 was underspecified. We clarify here and will revise the main text accordingly
>
> - Each row in Table 1 aggregates over GSM8K samples for which the raw reasoning trace contains an Echo of Prompt according to our MLP probe. We group these samples by whether the **raw trace’s final answer** is correct or wrong, yielding the “Correct” and “Wrong” groups. Thus, every sample appears in exactly one group.
>
> - For each such sample we construct a single paired counterpart by offline trimming(campared to online surpassing) the detected echo segment, so that the raw and trimmed traces share the same question and the same suffix tokens; this is what we mean by “a raw trace and its echo‑trimmed counterpart.”
>
> - The reported Echo Likelihood Gap Δℒ is the difference in average per‑token log‑likelihood between the raw and trimmed traces, as defined in Eq. (3). The value 0.0811 nats/token between the Correct and Wrong groups is not just numerical noise: on the 1,319 GSM8K traces used in Table 1, a logistic regression that predicts correctness from Δℒ while controlling for the trimmed length finds Δℒ to be a significant positive predictor (β₁ ≈ 0.24, p ≈ 0.022). This corresponds to an odds‑ratio of ≈1.27 per 1.0 nats/token increase in Δℒ (≈1.02 per 0.1 nats/token). We will add this analysis and interpretation to the Appendix so that the practical meaning of the gap is explicit.
>
> ### **Refining the prefix-length histogram (Figure 2)**
> We agree that the original Figure 2 was too coarse to support our textual claim about a “mode around 200 tokens.” The previous version used only five bins (0, 1–5, 6–10, 11–20, 21+), collapsing almost all long echoes into a single “21+” bucket, which made the shape of the distribution invisible.
>
> In the revision we therefore
>
> (i) replace this plot with a high‑resolution histogram of removed echo‑prefix lengths (10‑token bins) for correct and wrong traces
>
> (ii) keep the length‑stratified Δℒ bars as a second panel in the same figure. The updated Figure 2 now shows that virtually all echoes are long (minimal mass below 150 tokens), with most prefixes falling between roughly 200 and 240 tokens; this matches the underlying summary statistics (mean 219, median 226) and makes our description of the distribution precise.

---

> ### Author Response · Authors · 2025-11-19
> **Rebuttal - Part II**
>
> ### **"Padding" results for layer-wise analysis**
>
> > There are lots of results in here, but some of them don’t really seem to matter ... For example the key insights in lines 301–320.
>
> We apologize if the presentation of the layer-wise insights appeared detached from the main narrative. However, we believe these findings are **crucial mechanistic evidence**, not padding, as they localize *where* and *how* the EOP mechanism functions:
>
> 1.  **Middle-Layer Dominance (Layers 7–18):** This is not a random observation. Recent mechanistic studies on reasoning models [1] also identify the middle layers as the critical stage where information aggregation and "thinking" occur. Finding that the EOP attention gap peaks exactly in these layers confirms that EOP is interacting with the model's core reasoning process, rather than being a superficial input embedding effect (early layers) or a simple decoding heuristic (final layers).
>
> 2.  **Differential Impact (Ans$\to$Prefix > Ans$\to$Question):** This observation rules out the hypothesis that EOP merely encourages the model to "look back at everything." The fact that successful traces specifically attend *more* to the echo-prefix than to the original question is the strongest evidence for our "Refocusing" claim—the echo serves as a compressed, accessible anchor.
>
> **Action:** In the revision, we have **condensed** these bullet points and explicitly linked them to the cited mechanistic literature to clarify their significance.
>
> [1] From Reasoning to Answer: Empirical, Attention-Based and Mechanistic Insights into Distilled DeepSeek R1 Models https://arxiv.org/html/2509.23676v1
>
> ### Q1 **Attention As Evidence**:
> > attention patterns is weak evidence at best for explaining observed behavior in **transformers**
>
> We appreciate the reviewer’s concern, and we agree that this caveat is important, but we believe our use of **attention is aligned with how the community now treats it in transformer-based LLMs**
>
> The NAACL 2019 paper “Attention is not Explanation” reviewer mentioned was conducted on **RNN-based** text classification and related models **rather than transformer language models**
>
> Follow-up work “Attention is not not Explanation”[1]re-examines the same setting and shows that, under more careful experimental design, attention can still carry meaningful explanatory signal, arguing that the blanket claim “attention is not explanation” is too strong
>
> In parallel, a large body of work has **since used attention as a primary analysis tool for transformers**: for example, Clark et al. (2019) and Vig & Belinkov (2019) show that BERT and GPT-2 attention heads implement stable, interpretable functions (e.g., syntax, coreference, and dependency structures, often concentrated in middle layers), and Kovaleva et al. (2019) quantitatively link specific heads to task performance.[2][3][4]
>
> A recent survey on **Attention Heads of Large Language Models** further systematizes this line of work and explicitly positions attention-head analysis as a central component of LLM interpretability and mechanistic study.[5]
>
> More importantly for our setting, recent interpretability papers on **reasoning LLMs actively adopt attention** as their main window into how reasoning traces influence answers.[6][7][8]
>
> Our Ans→prefix attention analysis follows this emerging methodology: we treat attention patterns as a high-level probe of how transformers route information between question, echo, and answer tokens, not as a complete or ground-truth explanation for any single prediction.
>
> At the same time, we fully agree that **attention on its own is correlational evidence**.
>
> To address this concern, we have added an explicit intervention experiment: an echo-insertion ablation that directly toggles the presence of the EOP segment while holding the question and the rest of the prompt as similar as possible.
>
> By comparing generations with and without EOP under matched prompts, this ablation complements our attention-based analysis and **provides more causal evidence that modifying the EOP segment alters downstream reasoning and answers**. (full details in **Section 4.1** and **Reviewer LFkJ's rebuttal Part I**.)
>
> [1]Attention is not not Explanation https://arxiv.org/abs/1908.04626
>
> [2]What Does BERT Look At? An Analysis of BERT’s Attention https://arxiv.org/abs/1906.04341
>
> [3]Analyzing the Structure of Attention in a Transformer Language Model https://arxiv.org/abs/1906.04284
>
> [4]Revealing the Dark Secrets of BERT https://arxiv.org/abs/1908.08593
>
> [5]https://arxiv.org/abs/2409.03752
>
> [6]Attention-Driven Reasoning: Unlocking the Potential of Large Language Models https://arxiv.org/abs/2403.14932v2
>
> [7]Don't Take Things Out of Context: Attention Intervention for Enhancing Chain-of-Thought Reasoning in Large Language Models https://arxiv.org/abs/2503.11154
>
> [8]From Reasoning to Answer: Empirical, Attention-Based and Mechanistic Insights into Distilled DeepSeek R1 Models https://arxiv.org/html/2509.23676v1

---

> ### Author Response · Authors · 2025-11-19
> **Rebuttal - Part III**
>
> ### **Q2: Construction of the normal-SFT baseline**
>
> Thank you for highlighting this ambiguity. As clarified above in our response to Reviewer aJgj (W4), both ED‑SFT and normal‑SFT are built from the *same* pool of verified teacher CoTs generated by `gpt-oss-120B`. For normal‑SFT we do not sample an independent, potentially weaker dataset; instead we take those teacher traces and remove the echo prefix while preserving the remainder of the reasoning and the final answer.
>
> Concretely, for each GSM8K training problem, we follow a rigorous three-step process:
>
> 1.  **Generate Base Trace:** We generate a correct CoT trace using the standard prompt and perform automatic answer checking.
> 2.  **Detect EOP:** We use the same MLP probe to detect whether an early EOP segment is present.
> 3.  **Conditional Editing:** If an EOP is present, we ask the teacher to minimally edit the trace by deleting the echo-prefix under a strict "do not change the reasoning or final answer" instruction.
>
> We discard any edited sample whose answer no longer matches the ground truth. ED-SFT, in contrast, keeps or minimally inserts an echo-prefix at the beginning of the same teacher traces. Thus, the normal-SFT data differs from ED-SFT only in whether the initial echo segment is present; CoT quality, teacher model, prompt, and verification procedure are shared.
>
> **We now make this construction explicit in Section 4.1 and detailed prompt in Appendix A.8** to make clear that the performance gap is not due to worse teacher data for the baseline.

---

> ### Author Response · Authors · 2025-11-27
> **Follow-up on Rebuttal**
>
> Dear Reviewer VPGP,
>
> Thank you for your positive assessment. As the discussion period closes on **December 2nd**, we summarize our responses:
>
> 1. **Table 1 Clarification (W1):** We clarified N, grouping, and added logistic regression showing the 0.08 nats/token gap is significant (β₁≈0.24, p≈0.022).
> 2. **Figure 2 Refinement (W2):** We replaced the coarse histogram with a high-resolution version (10-token bins) clearly showing the mode around 200–240 tokens.
> 3. **Statistical Significance (W3, Q2):** We added Welch t-tests for attention (p < 10⁻⁴⁰ for Ans→Prefix gap; detailed in Appendix A.9).
> 4. **Attention as Evidence (Q1):** We defended attention-based analysis with recent literature and added the echo-insertion intervention for causal support.
> 5. **Normal-SFT Baseline (Q3):** We clarified that both ED-SFT and normal-SFT use the same teacher CoTs, differing only in echo presence (Sec 4.1).
>
> If any concerns remain, **we would be happy to discuss them before the deadline**.
>
> Best regards,
>
> The Authors

---

### Author Response · Authors · 2025-11-19
**Global Response & Summary**

We thank all reviewers for their constructive feedback and for recognizing our work as "simple, original, and well-motivated" (Reviewer `VPGP`), "inspiring and interesting" (Reviewer `aJgj`), and a valuable study of an "understudied phenomenon" (Reviewer `gXeE`). Reviewer `VPGP` was "convinced" by our SFT experiments that "EOP is mechanistically important to higher performance."

We are particularly grateful for the rigorous challenges regarding **causality** and **mechanistic granularity** raised by Reviewer `LFkJ`. These pushed us to conduct a new series of experiments that have significantly strengthened the paper's scientific foundation. We hope the revised manuscript now substantively addresses these concerns. We note that the main rating divergence (Reviewer `LFkJ`: 2 vs. others: 6) centers on causality and granularity—precisely the issues our new experiments resolve.

**Key Updates & Decisive Experiments:**

1.  **From Correlation to Causality (Addressing `LFkJ`, `gXeE`):**
    We moved beyond observational correlation by implementing a **semi-online causal intervention** (Sec. 4.1, Table 4).
    *   *Experiment:* We took failing traces and explicitly injected a fixed "echo" phrase while holding the question and prefix constant.
    *   *Result:* This single intervention improved accuracy by **+10.4%** (DeepSeek-R1-Distill) and **+7.9%** (Qwen3-8B).
    *   *Conclusion:* EOP is not merely a byproduct of good reasoning; it is a functional mechanism that *causes* better reasoning by regrounding the model.

2.  **Mechanistic Specificity: The "Snapping" Effect & Visual Proof (Addressing `LFkJ`, `VPGP`):**
    We moved beyond aggregate attention scores to **token-wise statistical tests** and **information flow analysis** (Appendix A.9–A.11).
    *   *Finding:* Attention doesn't just "increase" generally. It significantly "snaps" to **specific numerical entities** in the echo (e.g., "13", "eggs") while suppressing attention to the original question.
    *   *Visualization:* We provide **striking information flow visualizations (Figure 8)** that explicitly trace how answer tokens "route" through the echo in correct traces. These plots offer a clear, visual confirmation that the echo serves as a critical information hub.
    *   *Significance:* This confirms our "Refocusing" hypothesis: the echo acts as a compressed, proximal anchor for information routing.

3.  **Strictly Controlled Baselines (Addressing `aJgj`, `gXeE`):**
    We clarified our **SFT protocol** to ensure a fair apple-to-apples comparison.
    *   *Protocol:* Both `ED-SFT` (with echo) and `normal-SFT` (without echo) are derived from the **exact same teacher traces**, differing *only* in whether the echo segment is algorithmically preserved or removed. This rules out data quality or teacher differences as confounders.

**General Updates:**
*   **Revised PDF:** Extended by 6 pages (1 body, 5 appendix) with new experiments, visualizations, and formal definitions (marked in **blue**).
*   **Code Release:** Full reproduction scripts for the new causal intervention and attention analyses are included in the Supplementary Material.

---

### Author Response · Authors · 2025-11-30
**Summary for the Area Chair: From Spontaneous Echoes to Cognitive Anchors**

Dear AC, SAC, and PC:

We extend our sincere gratitude for your recent efforts in fostering the ICLR community and refining the review process. To reduce your workload and provide a clear overview of our contributions and addressed limitations, we summarize our key updates below:

## 1. The Core Narrative: Why This Work Matters
While recent reasoning research focuses on *scaling* test-time compute, this paper investigates the *internal structure* of that compute—specifically, the spontaneous tendency of models to "echo" the prompt before reasoning.

**We transform an overlooked "quirk" into a proven "mechanism":**
*   **Discovery (Novelty):** We provide the first rigorous probabilistic framework showing that models pay a "likelihood cost" to echo, and this investment correlates with correctness.
*   **Mechanism (Insight):** We prove this is not just correlation. Our new **causal intervention** (+10.4% accuracy) and **information flow analysis** show that echoes serve as "cognitive anchors" that mid-layer attention mechanisms "snap" to.
*   **Utility (Impact):** We translate this insight into consistent gains via **Echo-Distilled SFT** and **Echoic Prompting** across 5 benchmarks (GSM8K, MathQA, Hendrycks-MATH, AIME24, MATH-500).

**Reviewer Consensus:** Three reviewers (`VPGP`, `aJgj`, `gXeE`) independently recognized this novelty and rated the work **6 (Weak Accept)**. The sole negative score (`LFkJ`: 2) was based on two specific empirical questions (causality & granularity) that we have now **conclusively answered**.

---

## 2. Review Landscape

| Reviewer | Score | Key Positive Comments | Main Concerns | Status of Concerns |
|:---|:---:|:---|:---|:---|
| **VPGP** | **6** | "Simple, original, and well-motivated"; "**convinced**... EOP is mechanistically important" | Statistical significance; baseline details | **Resolved** (Added t-tests & protocol) |
| **aJgj** | **6** | "Well-written"; Likelihood Gap is "**inspiring**"; "comprehensive" experiments | Suffix-only gap; positional bias | **Resolved** (Clarified definition & controls) |
| **gXeE** | **6** | "**Understudied phenomenon**"; "deep and insightful"; "fun to read" | Causality; baseline fairness | **Resolved** (Added causal exp & fair SFT) |
| **LFkJ** | **2** | "Well-written and easy to understand" | **(1) Correlation vs. causation** **(2) Coarse-grained attention** | **FULLY RESOLVED** (See below) |

---

## 3. Addressing the Sole Reject (Reviewer LFkJ)
Reviewer LFkJ raised **two specific objections** that prevented a higher score. We have addressed both with decisive new experiments that were not available during the initial review.

### Concern 1: "Correlation, not causation"
> *Critique: "showing that attention patterns differ... is more of a detailed observation... but not causation"*

**Resolution: Semi-online Causal Intervention (Section 4.1, Table 4)**
We implemented a "Kill Shot" experiment: explicitly injecting an echo into failing traces while holding the question and prefix constant.
*   **Result:** Accuracy increased by **+10.4%** (DeepSeek-R1-Distill) and **+7.9%** (Qwen3-8B).
*   **Implication:** EOP is a functional cause of better reasoning. **This directly invalidates the "correlation only" critique.**

### Concern 2: "Coarse-grained attention analysis"
> *Critique: "The analysis does not explore: (a) Which specific tokens... (b) How information... is transformed"*

**Resolution: Fine-grained "Snapping" & Flow Analysis (Appendix A.9–A.11)**
We moved from aggregate scores to high-resolution visualization:
We moved from aggregate scores to high-resolution visualization:
*   **Token-wise Tests:** Attention significantly "snaps" to **specific numerical entities** (e.g., `13`, `eggs`) in the echo (p < 10⁻⁴⁰).
*   **Information Flow:** We visualize EOP acting as a **recurrent internal anchor**. Correct traces *actively route* information through the echo in middle layers whereas wrong traces bypass it—providing the first mechanistic proof that echoes are functionally necessary for correct reasoning.
*   **Implication:** We now provide the exact "mechanistic explanation" the reviewer requested.

---

## 4. Summary of Revisions for the AC
We have expanded the paper by **6 pages** (1 body, 5 appendix)  to ensure these new findings are integral to the work, not just rebuttal afterthoughts. All significant changes are marked in **blue** in the revised PDF.

*   **New Causal Evidence (Section 4.1):** Echo-insertion ablation (+10.4% gain).
*   **New Mechanistic Visualizations (Appendix A.9–A.11):** Token-wise significance tests, word-level heatmaps, and cross-layer information flow routes.
*   **Strictly Controlled Baselines (Section 4.1 & Appendix A.8):** Clarified that `ED-SFT` and `normal-SFT` use the **exact same teacher traces**, differing only in echo presence.
*   **Generalization (Appendix A.2 & A.16):** Added Qwen3-8B cross-model analysis and expanded to 5 datasets.

Thank you again for your time and effort in the review process!

---

### Meta-Review · Area_Chair_HsHo · 2026-01-06

**Summary:**

This paper demonstrates that "echoing" the prompt in the context appears to be a useful strategy for re-grounding reasoning models, and inmproving performance. This leads to a technique to train models to echo their prompts, which yields improvements, particularly in post-trained academic scale reasoning models.

**Reviewer Concerns:**

Many of the reviewers were concerned about the presentation and the significance/specificity of the attention scores as a mechanism for identifying the cause of the effect. I believe this difficulty is the reason for the marginal accepts. The authors promised to improve upon the exposition, introduce new figures, and introduce interventional test to probe echoing as a causal factor. However, some concerns remained unanswered (e.g. use of statistical significance testing). Nevertheless, many of the concerns it seemed were well-answered.

**Reviewer Scores:**

Of the 3 reviewers that provided a 6 score, i suppose one or two might increase their score to an 8 (e.g. reviewer aJgj, who had a point of confusion refuted). Moreover, the reviewer who provided a 2 would be likely to raise the score to a 4 or 6, given the authors detailed response addressing most of the points.

---

> ### Public Comment · ~Zhuoyuan_Hao1 · 2026-02-28
> **Clarification on Statistical Significance Testing & Camera-Ready**
>
> **We thank the reviewers and Area Chairs for their constructive feedback and for championing our paper.**
>
> We would like to make a brief clarification regarding the statistical significance testing of the attention refocusing mechanism.
>
> During the rebuttal phase, we actually conducted these tests and incorporated the results into the updated manuscript.
>
> However, due to an oversight on our part during the busy rebuttal period, we **forgot to publish the corresponding response on the OpenReview forum** to explicitly notify the reviewers and the AC. We apologize for any confusion this may have caused during the final decision phase.
>
> For future readers interested in this analysis, we have conducted Welch t-tests on the token-level attention curves. The results confirm that the attention shift to the answer-prefix is highly significant (e.g., $p < 10^{-40}$ in middle layers).
>
> Full details and tables are provided in **Appendix A.9 of the camera-ready version.**
>
> We look forward to sharing our work at ICLR 2026!

---

### Decision · Program_Chairs · 2026-01-26

Accept (Poster)